# Integrated Development of a Topology-Optimized Compliant Mechanism for Precise Positioning

Yaoyuan Hu , Bingfeng Ju and Wule Zhu *

The State Key Laboratory of Fluid Power Transmission and Control, School of Mechanical Engineering, Zhejiang University, Hangzhou 310027, China; yyhu@zju.edu.cn (Y.H.); mbfju@zju.edu.cn (B.J.)
* Correspondence: wulezhu@zju.edu.cn

**Abstract:** A scheme for modelling and controlling a two-dimensional positioning system with a topology-optimized compliant mechanism is presented. The system is designed to ensure a relatively large workspace and exhibit robustness against system nonlinearities. A detailed design procedure based on topology optimization is presented, and a nonlinear description of the designed mechanism is developed as a starting point for further precise position control. The theoretical model is shown to be suitable for a considerably larger working range without losing consistency. A backstepping controller is employed to manipulate the nonlinearities in the model resulting from the geometrical and material nonlinearity of the mechanical structure. The hysteresis of the piezoelectric actuator is also taken into consideration. An experimental verification of the controller demonstrates that the proposed design approach improves the performance of compliant mechanism and satisfies the needs for precision positioning.

**Keywords:** topology optimization; backstepping robust control; system identification; compliant mechanism



## 1. Introduction

Compliant mechanisms [1,2] have been widely used over the last few decades as a promising routine for transforming motions [3], forces [4], or energy [5,6] from input to output. Unlike rigid-body mechanisms, compliant mechanisms are monolithic and gain their mobility from the deflection of flexible members rather than movable joints. This offers increased precision and reliability combined with reduced wear, eliminating the need for lubrication. Such systems are often designed with certain aims, not only at the macro-scale, such as large-input displacement amplification [7] or sufficiently high output stiffness [8,9], but also at the micro-scale, such as micro-electromechanical systems (MEMS) [10] and surgical applications [11]. As design problems become more complicated, the continuum topology optimization (TO) freeform design methodology has also become a popular routine for such mechanisms [4,12–16]. In TO, the traditional trial-and-error design approach is replaced by an automated iterative design approach, which determines the optimal material distribution in a finite-element model while minimizing a given cost function [17,18]. This allows for the design of components or systems based on prescribed loads and boundary conditions, which harness rigorous optimization and simulation schemes to achieve superior performance.

Piezoelectric-actuated compliant mechanisms composed of piezoelectric actuators (PEA) and TO compliant mechanisms (CM) are one of the promising applications in precision positioning, owing to the fast response and extreme positioning resolution of PEAs. These systems also gain flexibility in applications from the TO method because, theoretically, they can take arbitrary design domains and boundary conditions.

However, unlike traditional kinematics-based approaches [1], which allow the designer to apply knowledge and approximations developed for rigid-body mechanisms kinetically [19,20] or dynamically [21,22], the dynamics of topology-optimized CMs are usually

nonlinear and complicated. This drawback of the TO method, together with the inherent hysteresis nonlinearity of PEAs and system uncertainties, leads to challenges in controlling such topology-optimized CMs, consequently limiting their practical applications. Further, although researchers have made significant advancements in topology-optimized CM design with multiple conditions, constraints, or output ports to composite motion [23–25] or to generate paths [26], the design of an optimization objective for maximum output workspace, which is a crucial attribute for multi-axis positioning stages, remains undone.

In this study, we aim to develop an integrated effective design and control scheme for multi-DOF positioning stage based on topology-optimized compliance mechanisms with a novel TO objective function designed for large output workspace. Firstly, the TO process is performed based on a nonlinear finite-element analysis. Secondly, a reduced spring-mass model with nonlinear disturbances is developed to cover the dynamics of the designed CM. Then, a MIMO backstepping robust controller is employed to achieve a high trajectory tracking performance. The output motions are not deliberately decoupled in the TO process; rather, the CM gains the ability to composite motions by tangent space modeling of the controller. The controller design process starts from the bounded-input-bounded-output PEA systems, which then serve as an ideal input for a new controller that stabilizes the subsequent compliant mechanism, making up a robust backstepping controller [27]. An experiment verification process is also given afterwards. The resulting system is intended to greatly extend the capabilities of such TO compliant mechanisms.

## 2. Topology Optimization

The topology optimization process in this study is based on the solid isotropic material with penalization (SIMP) [28,29] approach. The basic idea is that each finite element is associated with a fictitious pseudo-density variable $\rho$, such that $0 \leq \rho \leq 1$, which parameterizes the topology optimization procedure. The general algorithm is shown in Algorithm 1, and in this section, we will follow the design steps.

---

**Algorithm 1 SIMP algorithm for topology optimization. SIMP Algorithm**

---

**Initialization:**
**Generate mesh and boundary conditions, define material properties, initialize pseudo-density**
**Main iteration loop:**
**While not convergent do**
**Finite element analysis**
**Objective and constraints evaluation**
**Update density**
**End while**
**Evaluation of results**

---

*2.1. Boundary Conditions and Model Specifications*

The first thing to do is to specify the boundary conditions of the design domain. Consider a standard compact setup from previous research [30,31] for a piezoelectric-actuated compliance mechanism as the input unit of the design domain, as shown in Figure 1a, and assume the specifications of the PEAs, as shown in Table 1. The total dynamics of a thus-designed PEA set can be described as a single-DOF spring–mass mechanical system, given by:

$$m_{in}\ddot{x}_{in} = F_{in} - f - F_{res}, \tag{1}$$

where $m_{in}$ is the mass of the output blocking plate; $F_{in}$ is the input force produced by the PEA unit; $F_{res}(x_{in}, \dot{x}_{in}) \approx K_{in}x_{in} + C_{in}\dot{x}_{in}$ is the lumped restraint force with $K_{in}$ and $C_{in}$ the spring stiffness and the damping coefficient of the flexure blocking plate, respectively; and $f$ is the output force of the total set, which is the opposite direction of the force to be

applied on the compliant mechanism. The force $F_{in}$ generated by the PEA can be modeled by a compressive spring according to the following:

$$F_{in} = K_{pea} \cdot \delta x, \tag{2}$$

where $K_{pea}$ is the stiffness of the PEA and $\delta x$ is the difference between the nominal load-free output $x_{nom}$ and the actual output $x_{in}$. Substituting Equation (2) into Equation (1) yields the total dynamic model of a single PEA set, as follows:

$$m_{in}\ddot{x}_{in} = K_{pea} \cdot (x_{nom} - x_{in}) - f - F_{res}(x_{in}, \dot{x}_{in}). \tag{3}$$

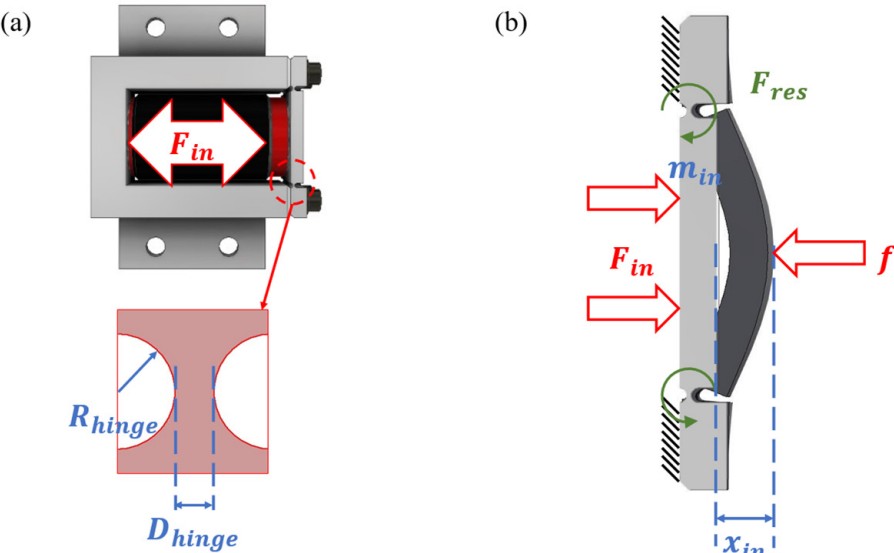

**Figure 1.** (**a**) Schematic illustration of a single PEA set; (**b**) Design parameters and simplified model of the output port.

**Table 1.** Specifications of the PEAs.

| Symbol | Quantity | Value |
|---|---|---|
| $K_{pea}$ | Stiffness | 250 [N/μm] |
| max $x_{nom}$ | Max stroke [a] | 60 [μm] |
| $f_{block}$ | Blocking force | $1.5 \times 10^4$ [N] |
| $R_{hinge}$ | Hinge radius | 0.75 [mm] |
| $D_{hinge}$ | Hinge thickness | 0.5 [mm] |

[a] The max stroke is the maximum of the nominal output $x_{nom}$ of a given PEA.

In this work, we focus on the study of a simplified 2-DOF positioning application. This process can be easily generalized to higher DOF and higher dimensional designs [32], and the basic ideas are similar. The whole setup is shown in Figure 2a.

The input forces applied to the design space are equal in magnitude but opposite in direction to the reaction force of the compliant mechanism. Because the two PEAs are fixed to the platform bed and the output of the PEAs are most likely to be horizontal, the inputs to the design domain can be assumed as forces parallel to the $x$-axis on roll supports, as shown in Figure 2b. In addition, we make the following assumptions concerning the input forces and the reaction forces ($f_i$s) of the compliant mechanism to simplify further discussions:

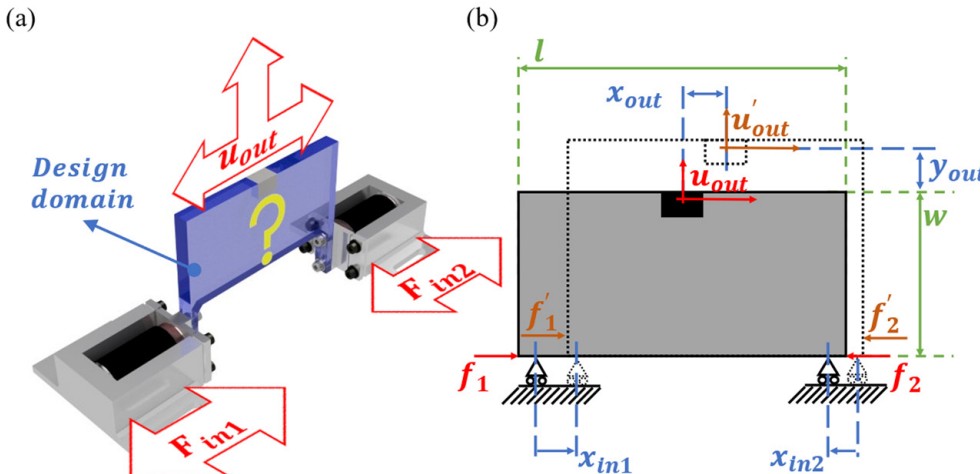

**Figure 2.** (**a**) Schematic illustration of a 2-DOF compliance set; (**b**) Simplification of the design domain and boundary conditions.

**Assumption 1.** *The input forces to the design domain are balanced by the internal residual forces at the corresponding input nodes.*

In other words, $f_i$s can be represented by the deformations and accelerations of the compliance mechanism. We will return to this assumption in the next subsection.

**Assumption 2.** *The resonance frequency of the design domain is much higher than the input frequency; therefore, the dynamics of the PEAs and the designed compliant mechanism can be separated without loss of generality.*

Suppose that the output vector $\boldsymbol{u}_{out}$ is decomposed into orthogonal vectors $\boldsymbol{x}_{out}$ and $\boldsymbol{y}_{out}$ along the *x*-axis and *y*-axis, respectively, and that the PEAs are almost identical with only minor differences. Typical working conditions, as shown in Table 2, summarize the relationship between the PEA stroke direction ($\rightarrow$ or $\leftarrow$, i.e., along or against the direction of the positive *x*-axis) and the output $\boldsymbol{y}_{out}$.

**Table 2.** Typical working conditions.

|  | Working Conditions | PEA1 Stroke $x_{in1}$ | PEA2 Stroke $x_{in2}$ |
|---|---|---|---|
| **Regular configuration** | Maximum $\boldsymbol{y}_{out}$ | $\rightarrow$ | $\leftarrow$ |
|  | Minimum $\boldsymbol{y}_{out}$ | $\leftarrow$ | $\rightarrow$ |
| **Reversed configuration** | Maximum $\boldsymbol{y}_{out}$ | $\leftarrow$ | $\rightarrow$ |
|  | Minimum $\boldsymbol{y}_{out}$ | $\rightarrow$ | $\leftarrow$ |

The reversed configuration assumes that $\boldsymbol{y}_{out}$ is along the negative *y*-axis under compressive inputs, useful when design domain width $w$ is limited.

### 2.2. Topology Optimization Process

Consider the 2D design problem of compliant mechanisms to maximize the positioning workspace, which occurs when the mechanism is used for precision positioning [33,34], and the whole process is performed in a static sense. The selection of optimization and material parameters is given in Table 3 [35].

**Table 3.** Algorithm and material specifications for the TO method.

| | | |
|---|---|---|
| | Penalty $p$ | **3** |
| **Optimization specifications** | Max iteration | 1000 |
| | Pseudo density lower bound $\rho_{\min}$ | $1 \times 10^{-6}$ |
| | Discretized steps $p_m$ | 40 |
| **Mesh specifications** | Length l | 12 [cm] |
| | Width w | 6 [cm] |
| | Thickness t | 1 [cm] |
| | Discretization | $n_l \times n_w = 120 \times 60$ |
| **Material properties** | Young's modulus E | 73 [GPa] |
| | Min modulus $E_{\min}$ [a] | 1 [kPa] |
| | Poisson ratio $\nu$ | 0.33 |
| | Material density $\rho_0$ | $2.71 \times 10^3$ [kg/m$^3$] |
| | Initial hyper $-$ elastic coefficient $c_1$ | $1 \times 10^{-5}$E [GPa] |
| | Hyper $-$ elastic strain threshold $\varepsilon^\star$ | 0.5 |

[a] A minimum modulus for the design domain material is assigned to void regions to prevent the stiffness matrix from becoming singular.

Suppose that the inputs given as in the previous section can be discretized into $p_m$ steps and can simultaneously satisfy the extreme working conditions listed in Table 2. Then, each pair of inputs at step $p$ yields a specific output vector, as shown in the schematic illustration in Figure 3. The aim of maximizing a reachable set of the mechanism is equivalently turned into maximizing the interior area bounded by the static trajectory, which is, in general, an irregular polygon. The area of the polygon is given by the $g$-determinant of the individual output vectors $u_{oj}$, $j = 1, 2, \ldots, p_m$ at each sampled step along the perimeter of the workspace, as the following [36]:

$$J = \text{gdet}(u_{o1}, \ldots, u_{opm}) = \frac{1}{2} \sum_{j}^{p_m} u_{oj} \times u_{oj+1}, \tag{4}$$

where the summation is cyclic, such that $u_{op_m+1} := u_{o1}$.

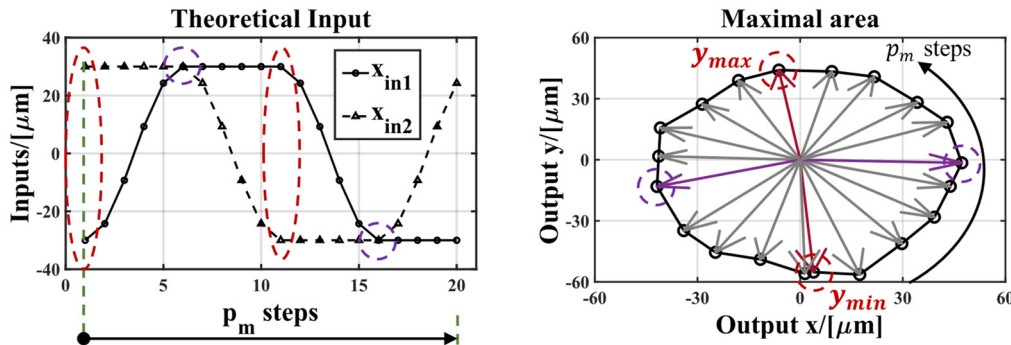

**Figure 3.** Proposed discretized input in this study and the corresponding $L_o$ output vectors on the perimeter of the workspace.

The constraints for the optimization process are imposed on both the input displacements and the total strain energy: the maximum stroke of the PEAs is limited to max $x_{in} = 60$ μm, and the stroke is smaller in practice because of the assembly error; the strain energy, $\Sigma$, is also constrained, with a coarse upper bound, $\Sigma_0$, in a manner similar

to that described in previous research [37,38] to avoid fatigue failures. As a result, the TO problem of maximizing $J$ can be formulated as follows:

$$\begin{aligned} \max_{\boldsymbol{\rho}} \quad & J \\ \text{s.t.} \quad & \sum_j^{p_m} \big| \boldsymbol{P}_j - \boldsymbol{R}_j\big(\boldsymbol{u}_j, \boldsymbol{\rho}_j\big) \big| = 0 \\ & \boldsymbol{\Sigma} \leq \boldsymbol{\Sigma}_0 \\ & u_{ini} < \max x_{nomi}, \quad i = 1,2 \\ & \boldsymbol{0} < \boldsymbol{\rho}_{min} \leq \boldsymbol{\rho} \leq \boldsymbol{1} \end{aligned} \quad . \tag{5}$$

In that the summation element $\big| \boldsymbol{P}_j - \boldsymbol{R}_j\big(\boldsymbol{u}_j, \boldsymbol{\rho}_j\big) \big|$ in the first constraint of the above Equation (5) is always positive, the following holds for all independent variables $\boldsymbol{u}_j$ and $\boldsymbol{\rho}_j$:

$$\boldsymbol{P}_j - \boldsymbol{R}_j\big(\boldsymbol{u}_j, \boldsymbol{\rho}_j\big) = 0. \tag{6}$$

The above Equation (6) is a nonlinear ordinary differential equation system for the unknown displacement field $\boldsymbol{u}$ and design parameter $\boldsymbol{\rho}$ within the calculation domain. The solution of this differential equation system is usually not analytical, and a conventional way is to approximate the displacement field by performing iterative Newton–Raphson method on the spatial discretization of the design domain [39,40]. A brief introduction of the nonlinear finite element synthesis including the additive hyper-elastic element modification to suppress instability during the optimization process [41] can be found in Appendix A.

Now consider the discretized finite element expression of the design domain. A vector $\boldsymbol{L}_o^T$ consisting of all zeros but 1 at the index of the output nodes can be used to select the desired output nodes from the discretized displacement field $\hat{\boldsymbol{u}}$, represented by:

$$\hat{\boldsymbol{u}}_o = \boldsymbol{L}_o^T \hat{\boldsymbol{u}}. \tag{7}$$

Since the following derivations will be mainly on this discretized representation, we will drop the hat representing the discretization process for simplicity. Thus, the sensitivity of the objective function can be represented by the summation of the sensitivities of the individual cross products, as follows:

$$\frac{\partial J}{\partial \rho_e} = \frac{1}{2} \sum_j^{p_m} \frac{\partial \big(\boldsymbol{u}_{oj} \times \boldsymbol{u}_{oj+1}\big)}{\partial \rho_e}. \tag{8}$$

By introducing a set of vectors of the Lagrangian multipliers $\lambda_j$, $j = 1, 2, \ldots, p_m$, for each output $\boldsymbol{u}_{oj}$, and assuming that the solution of Equation (6) has already been found via the Newton–Raphson method, the term $\lambda_j^T \big(\boldsymbol{R}_j(\boldsymbol{u}_j) - \boldsymbol{P}_j\big)$ is equal to zero and can be added to the displacement vector without changing the result, yielding:

$$\frac{\partial J_j}{\partial \rho_e} = \frac{1}{2} \frac{\partial\Big[\Big(\boldsymbol{u}_{oj} + \lambda_j^T\big(\boldsymbol{R}_j - \boldsymbol{P}_j\big)\Big) \times \Big(\boldsymbol{u}_{oj+1} + \lambda_{j+1}^T\big(\boldsymbol{R}_{j+1} - \boldsymbol{P}_{j+1}\big)\Big)\Big]}{\partial \rho_e}. \tag{9}$$

We define the total residual for the $j$th load step as follows:

$$\mathfrak{R}_j\big(\boldsymbol{u}_j, \boldsymbol{\rho}\big) = \boldsymbol{P}_j - \boldsymbol{R}_j\big(\boldsymbol{u}_j, \boldsymbol{\rho}\big). \tag{10}$$

Then, the sensitivity of this modified function can be given by:

$$\frac{\partial J_j}{\partial \rho_e} = \frac{1}{2}\left(\boldsymbol{L}_{oj}\frac{\partial \boldsymbol{u}_j}{\partial \rho_j} + \lambda_j^T\left(\frac{\partial \Re_j}{\partial \boldsymbol{u}_j}\frac{\mathrm{d}\boldsymbol{u}_j}{\mathrm{d}\rho_e} + \frac{\partial \Re_j}{\partial \rho_e}\right)\right) \times \left(\boldsymbol{u}_{oj+1} + \lambda_{j+1}^T\Re_{j+1}\right)$$
$$+ \frac{1}{2}(\boldsymbol{u}_{oj} + \lambda_j^T\Re_j) \times \left(\boldsymbol{L}_{oj+1}\frac{\partial \boldsymbol{u}_{j+1}}{\partial \rho_e} + \lambda_{j+1}^T\left(\frac{\partial \Re_{j+1}}{\partial \boldsymbol{u}_{j+1}}\frac{\mathrm{d}\boldsymbol{u}_{j+1}}{\mathrm{d}\rho_e} + \frac{\partial \Re_{j+1}}{\partial \rho_e}\right)\right). \tag{11}$$

in accordance with Ref. [42]. By doing some arithmetic and transformations, and introducing the tangent stiffness matrix $\boldsymbol{K}_T$ as the linearized approximation of $\frac{\partial \Re_j}{\partial \boldsymbol{u}_j}$, we have:

$$\frac{\partial J_j}{\partial \rho_e} = \frac{1}{2}\left(\left(\boldsymbol{L}_{oj} + \lambda_j^T\frac{\partial \Re_j}{\partial \boldsymbol{u}_j}\right)\frac{\mathrm{d}\boldsymbol{u}_j}{\mathrm{d}\rho_e} + \lambda_j^T\frac{\partial \Re_j}{\partial \rho_e}\right) \times (\boldsymbol{u}_{oj+1} + \lambda_{j+1}^T\Re_{j+1}) +$$
$$\frac{1}{2}(\boldsymbol{u}_{oj} + \lambda_j^T\Re_j) \times \left(\left(\boldsymbol{L}_{oj+1} + \lambda_{j+1}^T\frac{\partial \Re_{j+1}}{\partial \boldsymbol{u}_{j+1}}\right)\frac{\mathrm{d}\boldsymbol{u}_{j+1}}{\mathrm{d}\rho_e} + \lambda_{j+1}^T\frac{\partial \Re_{j+1}}{\partial \rho_e}\right)$$
$$= \frac{1}{2}\left((\boldsymbol{L}_{oj} + \lambda_j^T\boldsymbol{K}_{Tj})\frac{\mathrm{d}\boldsymbol{u}_j}{\mathrm{d}\rho_e} + \lambda_j^T\frac{\partial \Re_j}{\partial \rho_e}\right) \times \left(\boldsymbol{u}_{oj+1} + \lambda_{j+1}^T\Re_{j+1}\right) +$$
$$\frac{1}{2}\left(\boldsymbol{u}_{oj} + \lambda_j^T\Re_j\right) \times \left((\boldsymbol{L}_{oj+1} + \lambda_{j+1}^T\boldsymbol{K}_{Ti})\frac{\mathrm{d}\boldsymbol{u}_{j+1}}{\mathrm{d}\rho_e} + \lambda_{j+1}^T\frac{\partial \Re_{j+1}}{\partial \rho_e}\right) \tag{12}$$

Notice that the Lagrange multiplier $\lambda_i$s can be chosen freely, such that $\boldsymbol{L}_{oi} + \lambda_i^T\boldsymbol{K}_{Ti} = \boldsymbol{0}$ is easily satisfied, then the coefficient term before $\frac{\mathrm{d}\boldsymbol{u}_\bullet}{\mathrm{d}\rho_e}$ can be eliminated immediately. As a result, we can write the sensitivity of the optimization objective regarding working area $J$ in the following form:

$$\frac{\partial J}{\partial \rho_e} = \frac{1}{2}\sum_j^{p_m}\left(\lambda_j^T\frac{\partial \Re_j}{\partial \rho_e} \times \left(\boldsymbol{u}_{oj+1} + \lambda_{j+1}^T\Re_{j+1}\right) + \left(\boldsymbol{u}_{oj} + \lambda_j^T\Re_j\right) \times \lambda_{j+1}^T\frac{\partial \Re_{j+1}}{\partial \rho_e}\right). \tag{13}$$

Since the evaluation of the gradients only requires the accuracy of a tangent space approximation [37], the approximated $\frac{\partial \Re_j}{\partial \rho_e}$ can be given by the following:

$$\frac{\partial \Re_j}{\partial \rho_e} = -p\rho_e^{p-1}(E_0 - E_{min})\boldsymbol{K}_{Tj}\boldsymbol{u}_i. \tag{14}$$

The design parameter is updated using the multi-criterion method of asymptotes (MMA) [35,43], in which case the sensitivities of the constraint functions are also needed. In this study, the input displacement sensitivity is almost identical to the expression given by Equation (13) [17], except that the selecting vector is changed to $\boldsymbol{L}_i$, which is defined for the input nodes in a similar sense to Equation (7). The total strain energy is given by the following:

$$\Sigma = \boldsymbol{u}^T R(\boldsymbol{u}) \approx \rho^p\hat{\boldsymbol{u}}^T\boldsymbol{K}_T\hat{\boldsymbol{u}}. \tag{15}$$

The sensitivity of Equation (15) is approximated by its linear part in the tangent space, determined elementwise [44] as follows:

$$\frac{\partial \Sigma_e}{\partial \rho_e} = -p\rho_e^{p-1}\hat{\boldsymbol{u}}_e^T\boldsymbol{K}_{eT}\hat{\boldsymbol{u}}_e. \tag{16}$$

By performing the iterative optimization process as given in Algorithm 1, the results of the TO are given in Figure 4. The initial calculated boundary of the output workspace is indicated by the dark blue parallelograms, and the final optimization results are indicated by the red parallelograms. The area of the viable output region is enlarged by a factor of over two with respect to the initial configuration. This intermediate result gives us an initial perspective on how the designed compliant reacts to certain input loads and will serve as the start point of the order reduction and controller design process in the following sections.

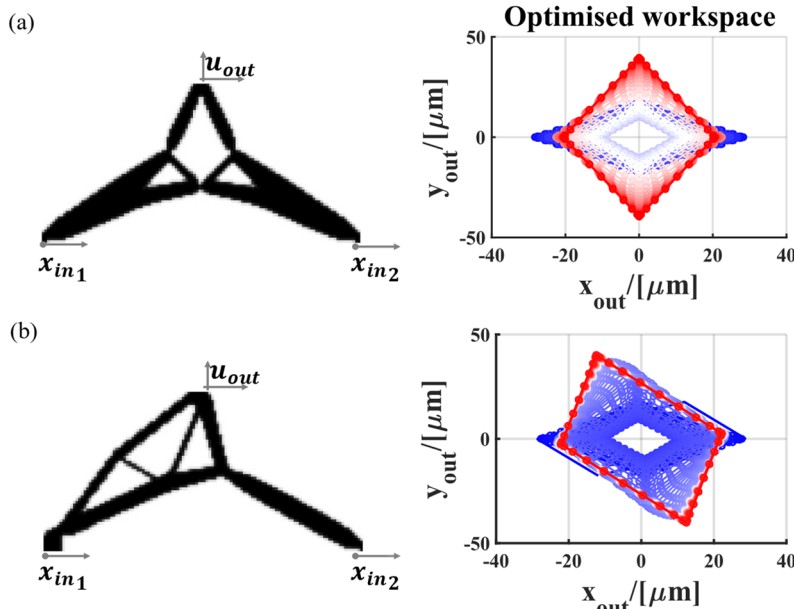

**Figure 4.** (**a**) TO results with extreme points set in accordance with the normal configuration in Table 2, and (**b**) TO results with a tilted output region w.r.t (**a**), and the calculated output workspace distribution evolution during the iterative optimization process.

## 3. System Modeling and Controller Design

The basic idea of the controller design for the whole positioning system can be outlined by simultaneously suppressing the influence of the PEA units and the nonlinearities of the designed CM. Considering the cascade nature of the system, i.e., the output displacement of a single PEA is first described as a reacting force on the output plate of the unit, which then serve as the input load to the CM, a robust backstepping controller is a suitable candidate for the controller structure design; by applying such controllers, the dynamics of the PEA and the passive CM part can be treated separately.

### 3.1. Inverse Multiplicative Compensation Scheme for PEA

By a careful selection of the compensation controller, the outputs of the PEA units can be seen as ideal displacement sources. Feedforward control is a common approach to compensate for the hysteresis effect [45,46]. The Prandtl–Ishlinskii, Duhem, and Bouc–Wen models were developed to describe the hysteresis effect, which is then used to construct the inverse controller for compensation of effects of hysteresis [47–50]. These nonlinear models motivated researchers to develop robust feedback controllers such as the sliding-mode controller [51,52], damping controller [53], disturbance observer [31], and adaptive robust controller [54,55]. In these control approaches, the hysteresis nonlinearity is treated as a disturbance to the tangent space systems. Specifically, in this section, an asymmetric Bouc–Wen model [49,56] is chosen to simulate the nonlinear hysteresis of the PEAs.

The Bouc–Wen model is based on an artificial state variable $h$. The model represents the hysteresis relationship between an excitation $F$ and the state $h$, according to the following differential equation:

$$\frac{\mathrm{d}h}{\mathrm{d}t} = A_{bw}\frac{\mathrm{d}F}{\mathrm{d}t} - B_{bw}\left|\frac{\mathrm{d}F}{\mathrm{d}t}\right|h|h|^{n-1} - \Gamma_{bw}\frac{\mathrm{d}F}{\mathrm{d}t}|h|^n + \delta_{bw}F\,\mathrm{sgn}\left(\frac{\mathrm{d}F}{\mathrm{d}t}\right),$$

$$\mathrm{sgn}(\cdot)\,\&= \begin{cases} 1, & x > 0 \\ 0, & x = 0 \\ -1, & x < 0 \end{cases},$$

(17)

where $A_{bw}$ is the amplitude of the restoring force, $B_{bw}$ and $\Gamma_{bw}$ control the shape of the hysteresis loop, $n$ controls the smoothness of the transition from elastic-to-plastic response, and $\delta$ is the non-symmetrical factor.

For PEAs, the excitation input $F$ in Equation (17) is replaced by the applied voltage $U$, and $n$ is set to 1 in accordance with the conventional practice. As a result, the Bouc–Wen model for PEAs can be expressed as follows:

$$\begin{cases} x_{nom}(t) = d_p U(t) - h(t) \\ \frac{\mathrm{d}h}{\mathrm{d}t} = A_{bw}\frac{\mathrm{d}U}{\mathrm{d}t} - B_{bw}\left|\frac{\mathrm{d}U}{\mathrm{d}t}\right|h - \Gamma_{bw}\frac{\mathrm{d}U}{\mathrm{d}t}|h| + \delta_{bw}U\,\mathrm{sgn}\left(\frac{\mathrm{d}U}{\mathrm{d}t}\right) \end{cases}, \tag{18}$$

where $x_{nom}$ is the displacement output. The parameter $d_p$ represents the piezoelectric coefficient and is strictly positive. A detailed parameter identification process of the involved parameters for the PEAs used in this study can be found in Appendix B; a list of the estimation results is also given.

The output of the PEA unit can be compensated separately following the inverse multiplicative scheme given by previous research [31,57,58]. Note that the state variable $h$ can be rewritten as:

$$h = H(U), \tag{19}$$

where $H(U)$ is a nonlinear operator characterized by the second equation in Equation (18). Then, the Bouc–Wen model can be reduced to the following:

$$x_{nom} = d_p U - H(U). \tag{20}$$

Suppose that we have a desired reference $x_{nom_d}$. Extracting the value of $U$ that meets the reference yields the following:

$$U = \frac{1}{d_p}\left(x_{nom_d} + H(U)\right). \tag{21}$$

An outline of the total inverse multiplicative compensation diagram is given in Figure 5. Thus, in the following designing process, the output of the PEA units will be simplified to ideal displacement sources with bounded disturbances and uncertainties, which can be addressed suitably by a robust controller.

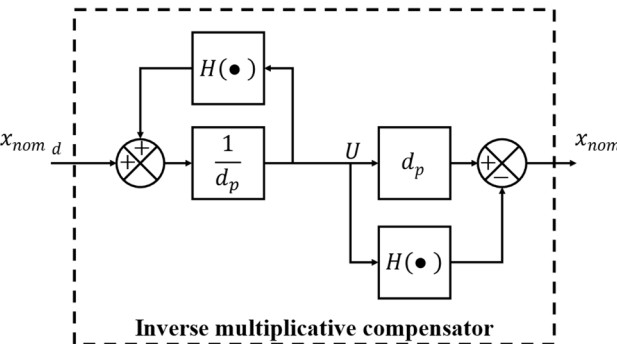

**Figure 5.** Inverse multiplicative hysteresis compensation diagram for the PEAs.

### 3.2. Controller Design

The controller design process is based on the total system dynamics considering both the PEA units and the TO designed compliant mechanism. Since the PEA units can be treated as ideal displacement sources with bounded disturbances, the discussion of this subsection will be focused on the dynamics of the compliant mechanism and the deduction of the actual form of the robust controller.

### 3.2.1. Reduced Order System Dynamics

Although there are handy controllers designed for finite element models considering fuzzy logic [57] or for set-invariance under uncertain constraints [58], it is more straightforward to consider the tangent space of the compliant mechanism and regard the bounded nonlinearities as disturbances and deviations.

Take the second result in Figure 4 as an example, we denote by $e = T(\theta) \begin{bmatrix} E_1 \\ E_2 \end{bmatrix}$ the local basis of a tilted output configuration with $T(\theta) = \begin{bmatrix} \cos\theta & \sin\theta \\ -\sin\theta & \cos\theta \end{bmatrix}$ the rotation matrix depicted in Figure 6a. We intuitively guess the geometric relation between the input displacement $x_{in} = \sum_i x_{ini} E_i$, the output compliant mechanism deformation $u_o = \sum_i u_{oi} E_i$, and the actual output $q = \sum_i q_i e_i$ in a static sense. The actual output is then represented as a vector sum of the moving spatial frame $x = \begin{bmatrix} \frac{1}{2}(x_{in1} + x_{in2}) \\ 0 \end{bmatrix}$ and the system deformation $u_o$ as follows:

$$q = x + u_o = \begin{bmatrix} \frac{1}{2}(x_{in1} + x_{in2}) + u_{o1} \\ u_{o2} \end{bmatrix}. \tag{22}$$

(a)  (b)

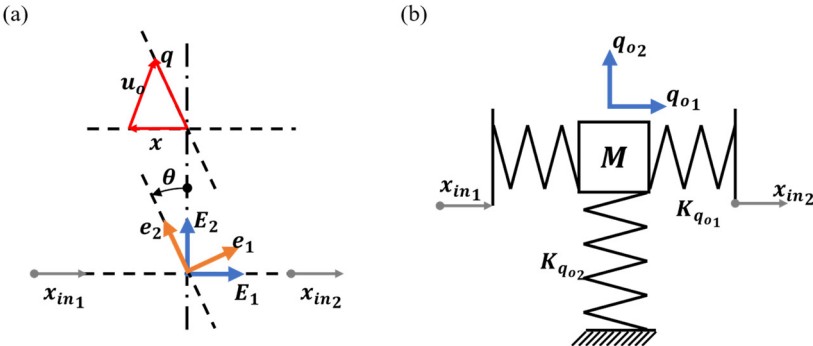

**Figure 6.** Schematic illustrations of (**a**) The 2nd-order spring–mass model of the compliant mechanism and (**b**) The tilted output configuration with rotation angle of θ.

Consider the geometric properties of the compliant mechanism setup, the output $q$ reaches its extreme value when the input boundary displacements satisfy $x_{in1} \approx -x_{in2}$. In this case, the theoretical output along the $x$-axis of the spatial frame also vanishes, and the output $u_o = [u_{o1}, u_{o2}]^T$ can be approximated by its discretized tangent space approximation in accordance with the solution of Equation (6). Consequently, the original load $P$ can also be replaced by a complementary displacement constraint [59,60], satisfying:

$$P = \begin{bmatrix} P_f \\ P_s \end{bmatrix} \approx \begin{bmatrix} K_{Tff} & K_{Tfs} \\ K_{Tsf} & K_{Tss} \end{bmatrix} \begin{bmatrix} u_f \\ u_s \end{bmatrix}, \tag{23}$$

where the matrices $K_{Tff}$, $K_{Tfs}$, $K_{Tsf}$, and $K_{Tss}$ are partitions of the original tangent stiffness $K_T$ introduced in Equation (11). Specifically, $K_{Tff}$ is the restrained structural stiffness matrix, which is square and symmetric, $K_{Tfs}$ and $K_{Tsf}$ are the off-diagonal submatrices relating to the known nodal displacement $u_s$ to the unknown reaction forces $P_f$ on the free nodes. Then, the output deformation of the compliant mechanism can be written as follows:

$$
\begin{aligned}
[u_o]^{2\times1} &= u_{fo} = L_o|_f u_f \\
&= \left[L_o|_f\right]^{2\times(N-2)} \left[(K_T)_{ff}^{-1}\right]^{(N-2)\times(N-2)} \left[K_{fs}\right]^{(N-2)\times2} [x_{in}]^{2\times1} \\
&= [\alpha]^{2\times2} x_{in},
\end{aligned}
\tag{24}
$$

where $L_o|_f$ is the selection vector given in Equation (7) restricted to the subset $u_f \subset u$,
$\boldsymbol{\alpha} = \begin{bmatrix} \alpha_{11} & \alpha_{12} \\ \alpha_{21} & \alpha_{22} \end{bmatrix}$ is the transfer matrix from the input to the output, $N$ is the total number
of finite elements of the design domain (including the elements with $\rho_e \approx 0$), and $s = 2$
corresponding to the two input displacements in our case. Thus, the elementary form of
Equation (24) is given as follows:

$$\begin{array}{l} u_{o1} = \alpha_{11} x_{in1} + \alpha_{12} x_{in2} \\ u_{o2} = \alpha_{21} x_{in1} + \alpha_{22} x_{in2} \end{array}. \tag{25}$$

We consider the following conditions:

i.    $x_{in1} = x_{in2}$, no compressive force exists, and $u_o = 0$;
ii.   $x_{in1} = -x_{in2}$, the spatial frame $x = 0$, and $u_o = q_o$.

When the output displacement reaches a maximum, the angle between $u_o$ and the
$y$-axis is presumed to be $\theta$, in which case $\frac{u_{o1}}{u_{o2}} = \tan \theta$. We then have the following two sets
of linear equations:

$$\begin{cases} \alpha_{11} x_{in1} + \alpha_{12} x_{in1} = 0 \\ \alpha_{21} x_{in1} + \alpha_{22} x_{in1} = 0 \end{cases} \Rightarrow \begin{cases} \alpha_{11} = -\alpha_{12} \\ \alpha_{21} = -\alpha_{22} \end{cases}$$
$$\begin{cases} \alpha_{11} \max x_{in1} - \alpha_{12} \max x_{in2} = q_{o1} \\ \alpha_{21} \max x_{in1} - \alpha_{22} \max x_{in2} = q_{o2} \end{cases} \Rightarrow \alpha_{11} = -\tan \theta \alpha_{22}. \tag{26}$$

This result indicates that the actual value of the matrix $[\boldsymbol{\alpha}]^{2\times2}$ can be represented by
a single value of $\alpha_{22}$. Thus, we can conclude the static relation between the input and the
output by an approximation of $\alpha_{22}$, which is given by the ratio of the maximum output and
the maximum compressive input as follows:

$$\alpha := \alpha_{22} \approx \frac{\max q_{o2}}{\max(x_{in1} - x_{in2})} = \frac{\max q_{o2}}{2 \max x_{in1}}, \tag{27}$$

yielding

$$\boldsymbol{\alpha}(\theta) \approx \begin{bmatrix} \tan \theta \alpha & -\tan \theta \alpha \\ \alpha & -\alpha \end{bmatrix}. \tag{28}$$

This result is especially useful when the full systematic model is unknown, so that one
can quickly provide an initial guess to the system's kinematic properties. We provide an
approximation of the actual output of our application in the following form:

$$q := \begin{bmatrix} q_1 \\ q_2 \end{bmatrix} = A(\theta) x_{in}, \tag{29}$$

where $A(\theta) = \begin{bmatrix} \frac{1}{2} + \tan \theta \alpha & \frac{1}{2} - \tan \theta \alpha \\ \alpha & -\alpha \end{bmatrix}$

When the system is working dynamically, i.e., tracking a certain output trajectory,
the whole compliant mechanism can be reduced to a dynamic spring–mass system in
correspondence with Equation (29) and Assumption 1, given by:

$$\begin{array}{l} M\ddot{q}_{o1} = 2K_{q_{o1}}(q_1 - q_{o1}) + 2C_{q_{o1}}(\dot{q}_1 - \dot{q}_{o1}) + \text{cross terms} \\ M\ddot{q}_{o2} = K_{q_{o2}}(q_2 - q_{o2}) + C_{q_{o2}}(\dot{q}_2 - \dot{q}_{o2}) + \text{cross terms} \end{array}, \tag{30}$$

where $K_{q_{oi}}$s and $C_{q_{oi}}$s are the lumped stiffness and damping coefficients of the spring–mass
system along the direction of $q_{oi}$, $i = 1, 2$, respectively, $M$ is the mass of the system, and
the cross terms denote the influence of the spring between the two output directions. An
illustration of this reduced system is given in Figure 6b. We can treat the cross terms
in Equation (30) as a disturbance term in the form of $\Delta_q = \begin{bmatrix} \delta_1(q_{o2}) \\ \delta_2(q_{o1}) \end{bmatrix}$, and we leave the

structural analysis of the cross terms for future research. For simplicity, we now consider the normal configuration, where $\tan\theta = 0$. The expansion of Equation (29) yields a direct tangent space approximation of the relation between the input $\boldsymbol{x}_{in}$ and the system output $\boldsymbol{q}_o$ as follows:

$$\boldsymbol{M}\ddot{\boldsymbol{q}}_o \& = \boldsymbol{\Delta}_q + \boldsymbol{K}(\boldsymbol{A}_0\boldsymbol{x}_{in} - \boldsymbol{q}_o) + \boldsymbol{C}(\boldsymbol{A}_0\dot{\boldsymbol{x}}_{in} - \dot{\boldsymbol{q}}_o), \tag{31}$$

where

$$\boldsymbol{M} = \begin{bmatrix} M & 0 \\ 0 & M \end{bmatrix}, \boldsymbol{K} = \begin{bmatrix} 2K_{q_{o1}} & 0 \\ 0 & K_{q_{o2}} \end{bmatrix}$$
$$\boldsymbol{C} = \begin{bmatrix} 2C_{q_{o1}} & 0 \\ 0 & C_{q_{o2}} \end{bmatrix}, \boldsymbol{A}_0 = \boldsymbol{A}(0) = \begin{bmatrix} \frac{1}{2} & \frac{1}{2} \\ \alpha & -\alpha \end{bmatrix}. \tag{32}$$

By including the developed model for the PEA units, we denote by $\boldsymbol{f}$ the interactive force between the inputs $x_{ini}, i = 1, 2$, and the output $\boldsymbol{q}_o$ as follows:

$$\boldsymbol{f} = \begin{bmatrix} f_1 \\ f_2 \end{bmatrix} = \begin{bmatrix} K_{q_{o1}}(x_{in1} - q_{o1}) + C_{q_{o1}}(\dot{x}_{in1} - \dot{q}_{o1}) \\ K_{q_{o1}}(x_{in2} - q_{o1}) + C_{q_{o2}}(\dot{x}_{in2} - \dot{q}_{o1}) \end{bmatrix}. \tag{33}$$

For the two PEA sets, we already have the matrix form of Equation (3):

$$\ddot{\boldsymbol{x}}_{in} = \boldsymbol{M}_{in}^{-1}\boldsymbol{K}_{pea}\boldsymbol{x}_{nom} - \boldsymbol{M}_{in}^{-1}\boldsymbol{K}_{in}\boldsymbol{x}_{in} - \boldsymbol{M}_{in}^{-1}\boldsymbol{C}_{in}\dot{\boldsymbol{x}}_{in} \\ + \boldsymbol{M}_{in}^{-1}\boldsymbol{K}_{in,q}\boldsymbol{q}_o + \boldsymbol{M}_{in}^{-1}\boldsymbol{C}_{in,q}\dot{\boldsymbol{q}}_o, \tag{34}$$

where

$$\boldsymbol{x}_{in} = \begin{bmatrix} x_{in1} \\ x_{in2} \end{bmatrix}, \boldsymbol{x}_{nom} = \begin{bmatrix} x_{nom1} \\ x_{nom2} \end{bmatrix}$$
$$\boldsymbol{M}_{in} = \begin{bmatrix} m_{in1} & 0 \\ 0 & m_{in2} \end{bmatrix}, \boldsymbol{K}_{pea} = \begin{bmatrix} K_{pea1} & 0 \\ 0 & K_{pea2} \end{bmatrix}$$
$$\boldsymbol{K}_{in} = \begin{bmatrix} K_{q_{o1}} + K_{in1} + K_{pea1} & 0 \\ 0 & K_{q_{o1}} + K_{in2} + K_{pea2} \end{bmatrix} \tag{35}$$
$$\boldsymbol{C}_{in} = \begin{bmatrix} C_{q_{o1}} + C_{in1} & 0 \\ 0 & C_{q_{o1}} + C_{in2} \end{bmatrix}$$
$$\boldsymbol{K}_{in,q} = \begin{bmatrix} K_{q_{o1}} & 0 \\ K_{q_{o1}} & 0 \end{bmatrix}, \boldsymbol{C}_{in,q} = \begin{bmatrix} C_{q_{o1}} & 0 \\ C_{q_{o1}} & 0 \end{bmatrix}$$

are the actual output vectors, mass matrix, stiffness matrices, and damping coefficient matrices, respectively.

Thus, the total system dynamics can be written as a combination of Equations (31) and (34), based on Assumption 2. Denoting the state variable vector by $\boldsymbol{z} = \begin{bmatrix} \boldsymbol{q}_o & \dot{\boldsymbol{q}}_o & \boldsymbol{x}_{in} & \dot{\boldsymbol{x}}_{in} \end{bmatrix}^T$ and the input vector $\boldsymbol{u} = \begin{bmatrix} x_{nom1} & x_{nom2} \end{bmatrix}^T$, we obtain the following linearized state-space form of the total system dynamics:

$$\dot{\boldsymbol{z}} = \boldsymbol{B}\boldsymbol{z} + \begin{bmatrix} \boldsymbol{0} \\ \boldsymbol{D} \end{bmatrix}\boldsymbol{u} + \boldsymbol{\Delta}, \tag{36}$$

where $\boldsymbol{B}$ and $\boldsymbol{D}$ are the corresponding system matrices, whose details can be found in Appendix C, and $\boldsymbol{\Delta} = \begin{bmatrix} \Delta_\eta & \Delta_\xi \end{bmatrix}^T$ is the model error, attributable mainly to the following:

i. finite-element modelling error $\epsilon_{FE}$
ii. piezoelectric modelling error $\epsilon_{PE}$
iii. asymmetric modelling error $\epsilon_{ASYM}$
iv. systematic noise $n_{SYS}$
v. measurement error $\delta_M$.

Additional uncertainties, such as assembly error and manufacturing defects, are also possible in practice. In this study, we only treat these errors as bounded (i.e., $\|\Delta\| \leq \delta_I$), matched, and Gaussian for simplicity.

### 3.2.2. Backstepping Robust Controller

In a practical setup, not all state variables in Equation (36) are measurable. In our case, specifically, only the output $q$ is measured. To build an explicit control scheme, estimates of the state variables are essential. Therefore, we utilize K-filters [27] to provide exponentially convergent estimates of the unmeasured states. We design the observer to obtain the estimate $\hat{z}$ as follows:

$$\dot{\hat{z}} = \mathbf{B}_0\hat{z} + k_{e1}z_1 + k_{e2}z_2 + bu + \Delta$$
$$y = \begin{bmatrix} z_1 & z_2 \end{bmatrix}, \tag{37}$$

where

$$\mathbf{B}_0 = \mathbf{B} - k_{e1}\underbrace{\begin{bmatrix} 1 & 0 & \cdots \end{bmatrix}}_{8} - k_{e2}\underbrace{\begin{bmatrix} 0 & 1 & 0 & \cdots \end{bmatrix}}_{8},$$

$$b = \begin{bmatrix} \mathbf{0} \\ D \end{bmatrix}. \tag{38}$$

By choosing a suitable $k_{e_d}$, we maintain the stability of observer matrix $\mathbf{B}_0$. Thus, there exists a symmetric and positive definite matrix $P$ such that:

$$P\mathbf{B}_0 + \mathbf{B}_0^T P = -\mathbf{I}, \quad P = P^T > \mathbf{0}. \tag{39}$$

Following the design procedure in [27,61], the K-filters are given by:

$$\hat{z} = \zeta + \Omega^T b, \tag{40}$$

where

$$\dot{\zeta} = \mathbf{B}_0\zeta + k_{e1}z_1 + k_{e2}z_2$$
$$\dot{\Omega}^T = \mathbf{B}_0\Omega^T + \begin{bmatrix} d_{11} & 0 \\ 0 & d_{22} \end{bmatrix}\begin{bmatrix} u_1e_7 \\ u_2e_8 \end{bmatrix}, \tag{41}$$

and where $e_i$ denotes the $i$th standard basis vector. The state-estimation error $\varepsilon = z - \hat{z}$ is readily shown to satisfy the following:

$$\dot{\varepsilon} = \mathbf{B}_0\varepsilon, \tag{42}$$

and will decay exponentially to zero.

Equation (33) can be modified into a state-space form representing a cascaded connection of two subsystems by selecting the state variables as $\eta = \begin{bmatrix} z_1 & z_2 & z_3 & z_4 & z_5 & z_6 \end{bmatrix}^T$ and $\dot{\xi} = \begin{bmatrix} z_7 & z_8 \end{bmatrix}^T$.

$$\dot{\eta} = \mathbf{B}_{\eta\eta} + \mathbf{B}_{\eta\xi}\xi + \Delta_\eta,$$
$$\ddot{\xi} = Du + \mathbf{B}_{\xi\eta}\eta + \mathbf{B}_{\xi\xi}\dot{\xi} + \Delta_\xi. \tag{43}$$

We use the following nonlinear input transformation:

$$u = D^{-1}\left[-\left(\hat{\mathbf{B}}_{\xi\eta}\hat{\eta} + \hat{\mathbf{B}}_{\xi\xi}\hat{\dot{\xi}}\right) + \dot{v}\right], \tag{44}$$

where $\hat{\mathbf{B}}_{\xi\eta}$ and $\hat{\mathbf{B}}_{\xi\xi}$ are the estimation of the coefficient matrices of $\mathbf{B}_{\xi\eta}$ and $\mathbf{B}_{\xi\xi}$ evaluated at states $\hat{\eta} = \begin{bmatrix} z_1 & z_2 & \hat{z}_3 & \hat{z}_4 & \hat{z}_5 & \hat{z}_6 \end{bmatrix}^T$ and $\hat{\dot{\xi}} = \begin{bmatrix} \hat{z}_7 & \hat{z}_8 \end{bmatrix}^T$. All the unmeasured state variables, i.e., $\hat{z}_i$s, are replaced by their estimations for the controller input calculations. Since the estimators are designed to converge asymptotically to the real value of the state variables, we will drop the hat above the state variables $\eta$ and $\dot{\xi}$ unless specifically mentioned. To reduce Equation (43) to the pure integrator from the new input $v$ to $\dot{\xi}$, such that:

$$\ddot{\xi} = v, \tag{45}$$

we must introduce an stabilizing control law $a(\eta)$ such that the subsystems in Equation (43) can be stabilized within finite time, as suggested in ref. [27]. The idea behind the selection of this intermediate control law is straightforward:

i. $\xi$ is the actual input of the first subsystem in Equation (43) and is controlled by $v$; $a$ is the desired input control law of the first subsystem in Equation (43). Therefore, if we can find some $v$ such that $\xi$ is able to track $a$ very closely, then the first subsystem in Equation (43) is automatically stabilized.

ii. The same for the second subsystem in Equation (43), where the problem turns into finding $u$ from the same $v$ based on the relation given in Equation (44).

Therefore, the ultimate goal is to find this specific $v$ satisfying both conditions. However, before that, we need to solve for the exact form of the ideal control law $a(\eta)$. We denote $\epsilon_i = \eta_i - \eta_{di} = z_i - z_{di}$, $i = 1, 2$, the tracking error between the output node $\eta_i = z_i$, and the desired trajectory $\eta_{di} = z_{di}$. The relative degree of the total system is 2 [62]. A selection of the feedback error term $s = \dot{\epsilon} + k_\epsilon \epsilon$ yields the total error dynamic in the form of:

$$\dot{s} = \begin{bmatrix} s_1\left(\eta, a(\eta), \dot{\hat{\xi}}, \eta_d\right) + \Delta_1 \\ s_2\left(\eta, a(\eta), \dot{\hat{\xi}}, \eta_d\right) + \Delta_2 \end{bmatrix}, \tag{46}$$

where $a(\eta)$ is the desired input for the subsystem in Equation (43), as discussed; the $\dot{\hat{\xi}}$ is used specifically to avoid ambiguity that considers the control Lyapunov function (CLF):

$$V(\eta) = \frac{1}{2}\begin{bmatrix} s_1^2 \\ s_2^2 \end{bmatrix}. \tag{47}$$

The path derivative of the CLF with respect to the solution $\eta(t)$ is as follows:

$$\dot{V}(\eta) = \begin{bmatrix} s_1 \dot{s}_1 \\ s_2 \dot{s}_2 \end{bmatrix}. \tag{48}$$

The desired controller output $a$ can be determined further as the combination of a linear feedback term $a_m$ that expresses exponential suppression of the tracking errors and a robust term $a_s$ to compensate for the systematic errors, in the form of:

$$a = a_m + a_s. \tag{49}$$

This kind of separation gives us additional flexibility to adjust the control law for better performance. Thus, the expression of $a$ can be deduced by solving the inequality assuming that $\dot{V}(\eta) < 0$ always holds in Equation (48), satisfying the Lyapunov stability criterion.

Note that there is also a deviation between the actual input $\xi$ and its desired value $a$; let $\epsilon_a$ be this deviation:

$$\epsilon_a = \xi - a. \tag{50}$$

Augmenting Equation (47) with a quadratic term of the error variable $\epsilon_a$, we obtain a CLF for the whole system:

$$V_\alpha(\eta, \xi) = V(\eta) + \frac{1}{2}\begin{bmatrix} \epsilon_{\alpha 1}^2 \\ \epsilon_{\alpha 2}^2 \end{bmatrix}. \tag{51}$$

The path derivative of $V_\alpha$ is computed as follows:

$$\dot{V}_\alpha = \dot{V}(\eta) + \begin{bmatrix} \epsilon_{\alpha 1}\dot{\epsilon}_{\alpha 1} \\ \epsilon_{\alpha 2}\dot{\epsilon}_{\alpha 2} \end{bmatrix}. \tag{52}$$

Again, we choose to separate the controller input $v$ into two parts, such that:

$$v = v_m + v_s. \tag{53}$$

where, same as Equation (49), $\boldsymbol{v}_m$ is the proportional feedback stabilizing controller input and $\boldsymbol{v}_s$ is a robust controller that eliminates the effects of the modelling errors. The expression can also be deduced by assuming that Equation (52) always holds. We denote the approximation of the calculation of the system input as $\hat{\boldsymbol{v}} = \boldsymbol{v}_m + \boldsymbol{v}_s$. The system input follows from Equation (44) and this completes the design of a backstepping robust controller for the simplified spring–mass model that was developed for the compliant mechanism. A schematic illustration of the whole closed-loop system with the backstepping robust control law is shown in Figure 7. A detailed deduction process for the controller design can be found in Appendix D.

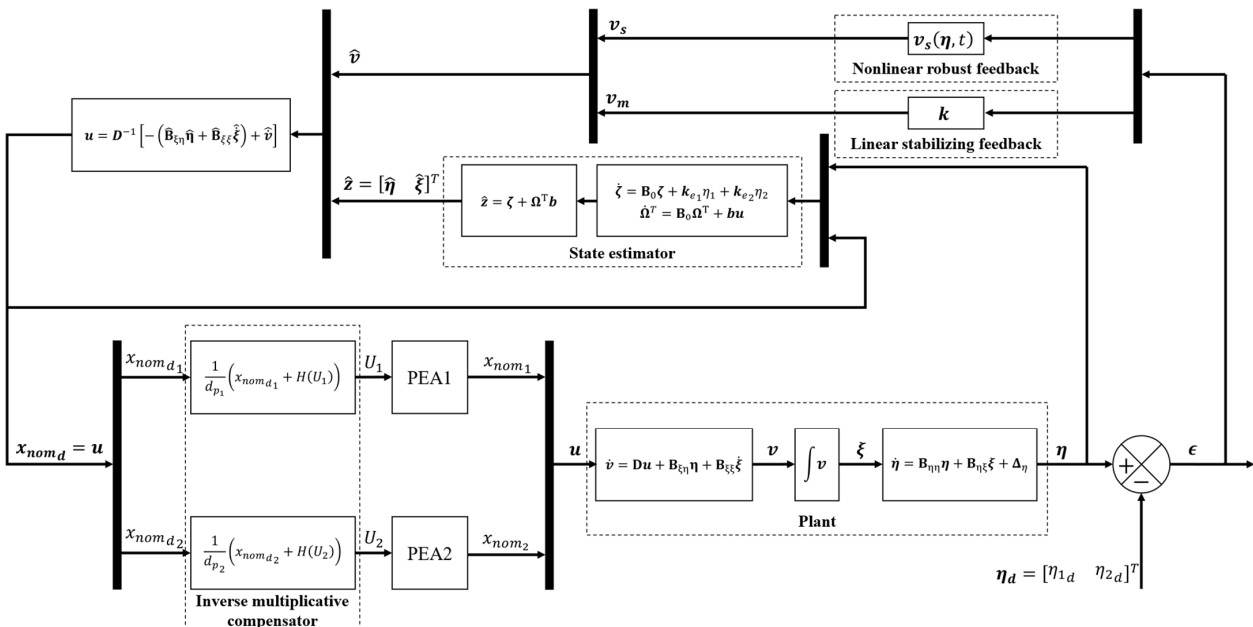

**Figure 7.** Schematic illustration of the total closed-loop system. The controller consists of a general state feedback controller to suppress the tracking errors and a robust controller to compensate for the nonlinearities occurring in the system.

## 4. Experimental Results

### 4.1. Experimental Setup

Experiments were carried out on a 2-DOF compliant setup, as shown in Figure 8. The compliant mechanism was manufactured using a wire-cutting method and made of aluminum alloy 6061. The PEA set output plates were made of spring steel. Initial guesses of the material property values are the same as those listed in Table 3. The adopted PEAs have a stroke of more than 60 [μm]. Other specifications are listed in Table 1. To accomplish the full cycle shown in Figure 4, the PEA sets were preloaded with a biased input voltage yielding a nominal output displacement of $\pm 30$ [μm]. The drivers of the PEAs were linear amplifiers (Type E-472.20, PI Inc.). The position sensors of the output were two capacitive distance sensors, one with a resolution of 7.5 [nm] and one with a resolution of 10 [nm]. The velocity signal was obtained from the difference between two consecutive position measurements, and the horizontal output displacement was derived by vector decomposition using the Pythagorean theorem. Note that in this setup the deformation of the joint will affect the precision of the measurement results, and so the control system is actually following a "nominal" trajectory with some minor differences to the actual desired displacement. This has little effect on a proof-of-principal experiment which mainly aims to show the efficiency of the control system. The measuring devices were connected to a data acquisition card via a noise-shielding I/O junction box with a sample time of less than 0.1 [μs]. The real-time codes of the control algorithm were explicitly implemented in computational software. The sampling period was set to 0.1[ms].

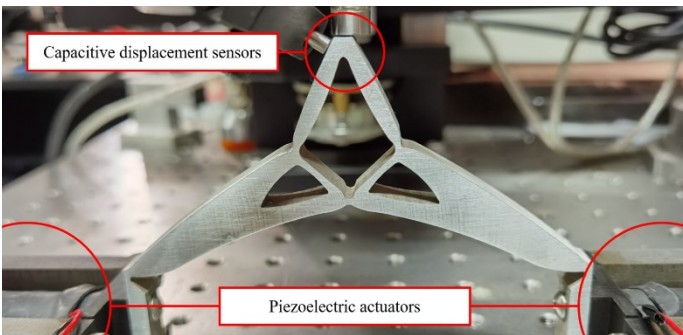

**Figure 8.** Image showing the experimental setup of the system. Two capacitance displacement sensors were used to measure the coupled output trajectory.

### 4.2. Experimental Results and Discussion

The values of the parameters of the system described by Equation (35) were estimated using an online least-squares estimator provided by the System Identification Toolbox in Simulink. The systems described by Equations (31) and (34) were discretized in time to make use of the estimators. The parameters to be evaluated, their corresponding regressors, and the reference outputs are given by the following:

$$\boldsymbol{\varphi}\& = \begin{bmatrix} \boldsymbol{A}_0\hat{\boldsymbol{x}}_{in}(t - T_s) - \boldsymbol{q}_o(t - T_s) \\ \boldsymbol{A}_0[\hat{\boldsymbol{x}}_{in}(t) - \hat{\boldsymbol{x}}_{in}(t - 2T_s)] - [\boldsymbol{q}_o(t) - \boldsymbol{q}_o(t - 2T_s)] \\ -\hat{\boldsymbol{x}}(t - T_s) \\ -[\hat{\boldsymbol{x}}_{in}(t) - \hat{\boldsymbol{x}}_{in}(t - 2T_s)] \end{bmatrix}, \tag{54}$$

$$\boldsymbol{\theta}\& = \begin{bmatrix} T_s^2\boldsymbol{M}^{-1}\boldsymbol{K} \\ \frac{T_s}{2}\boldsymbol{M}^{-1}\boldsymbol{C} \\ T_s^2\boldsymbol{M}_{in}^{-1}\boldsymbol{K}_{in} \\ \frac{T_s}{2}\boldsymbol{M}_{in}^{-1}\boldsymbol{C}_{in} \end{bmatrix}, \tag{55}$$

$$\boldsymbol{u}_r = \begin{bmatrix} \boldsymbol{q}_o(t) - 2\boldsymbol{q}_o(t - T_S) + \boldsymbol{q}_o(t - 2T_S) \\ \hat{\boldsymbol{x}}_{in}(t) - 2\hat{\boldsymbol{x}}_{in}(t - T_S) + \hat{\boldsymbol{x}}_{in}(t - 2T_S) - \boldsymbol{u} + \boldsymbol{M}_{in}^{-1}\boldsymbol{K}_{in}\boldsymbol{x}_{in}(t - T_S) \end{bmatrix}, \tag{56}$$

where $T_s$ represents the sampling period of the controller, and was set to $1 \times 10^{-4}$ [s] in our case. Note that some of the parameters were not evaluated. The masses were weighed on an electronic scale. The amplification ratio was approximated using the maximum as shown in Equation (27). The values of parameters that only appear in the additional terms, such as $K_{pea_i}$s in $\boldsymbol{K}_{in}$, were calculated algebraically from the estimated $K_{in_i}$s and $K_{qo_1}$. Initial guesses of the model parameters, as well as their final estimations, are provided in Table 4. The state estimator gain and feedback loop gain in the controller design process are also given in Table 4.

Lissajous curves with specified ratios of 3 and 2 were chosen as the desired output trajectories. Specifically, Figure 9a shows the time-domain performance of the designed system tracking a ratio-3 Lissajous curve without compensating for the nonlinearities by a robust controller, as a comparison. This is realized by aborting the robust feedback output in the block diagram of Figure 8. Figure 9b shows the performance of the complete system tracking the same desired ratio-3 Lissajous trajectory. The tracking of a ratio-2 curve is given in Figure 10a; a circular trajectory was also performed, as shown in Figure 10b. A two-dimensional plot for the space-domain trajectories is shown in Figure 11, for all four different experiments.

**Table 4.** Evaluated system parameters and controller design specifications.

| Part | Quantity | Guess | Estimate |
|---|---|---|---|
| **CM** | Horizontal stiffness $k_{qo1} \times 10^6$ [N/m] | 5 | 3.78 |
| | Vertical stiffness $k_{qo2} \times 10^6$ [N/m] | 5 | 4.03 |
| | Mass M [kg] | 0.046 | $-$ [a] |
| **PZT set** | Amplification ratio $\alpha$ | 1.012 | $-$ |
| | PZT1 stiffness $K_{pea1} \times 10^8$ [N/m] | 2.5 | $-$ |
| | PZT2 stiffness $K_{pea2} \times 10^8$ [N/m] | 2.5 | $-$ |
| | Input plate stiffness $K_{in1} \times 10^5$ [N/m] | 1 | 0.92 |
| | Input plate stiffness $K_{in2} \times 10^5$ [N/m] | 1 | 0.89 |
| | Input plate mass $M_{ini}$ [kg] | 0.042 | $-$ |
| **State Estimator** | Estimator gain $k_{e_1}$ | $1 \times 10^5$ | |
| | Estimator gain $k_{e_2}$ | | |
| **Controller design** [b] | Error gain $k_{\epsilon_1}$ | $2 \times 10^4$ | |
| | Error gain $k_{\epsilon_2}$ | $1.5 \times 10^4$ | |
| | Feedback gain $k_1$ | $1.2 \times 10^4$ | |
| | Feedback gain $k_2$ | $1.2 \times 10^3$ | |

[a] The masses in this table were measured on an electronic scale and were assumed to remain constant throughout the experiments. [b] The functionalities of the gains in the controller design can be found in Appendix C.

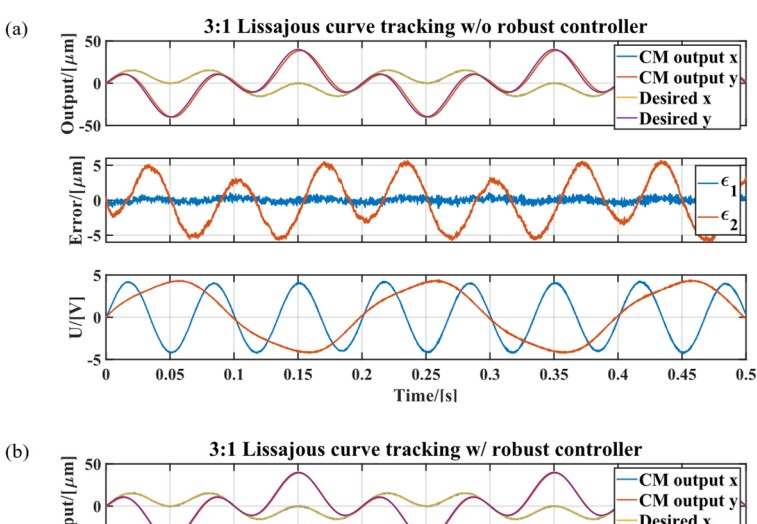

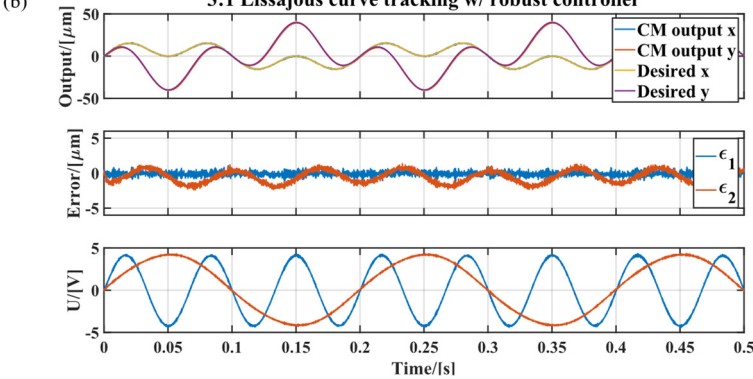

**Figure 9.** Experiment result showing a comparison between (**a**) Tracking of 3:1 Lissajous curve without the robust controller and (**b**) Tracking of 3:1 Lissajous curve with the robust controller.

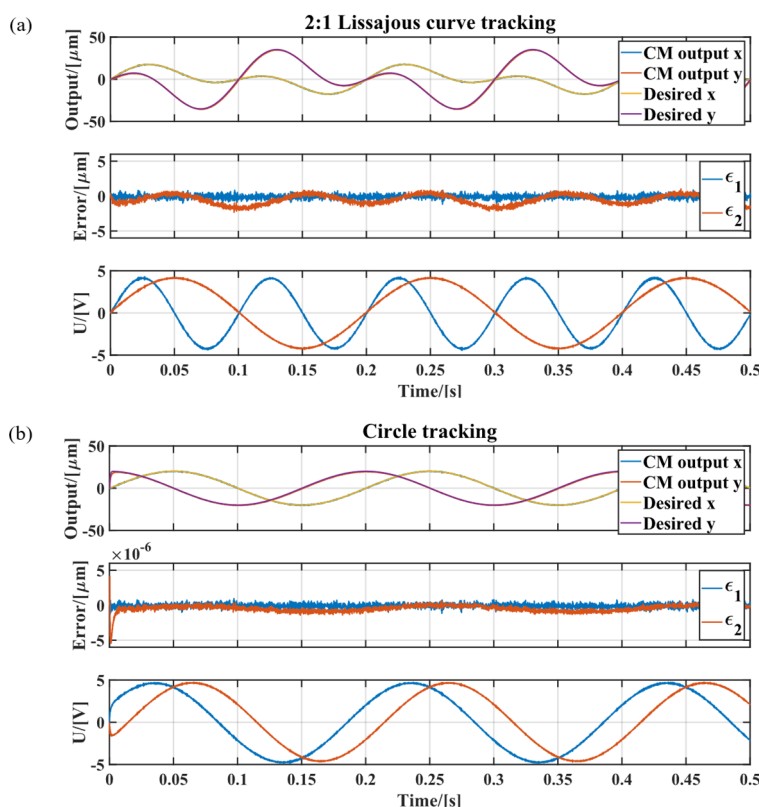

**Figure 10.** Experiment result showing the CM tracking (**a**) A 2:1 Lissajous curve and (**b**) A circular trajectory.

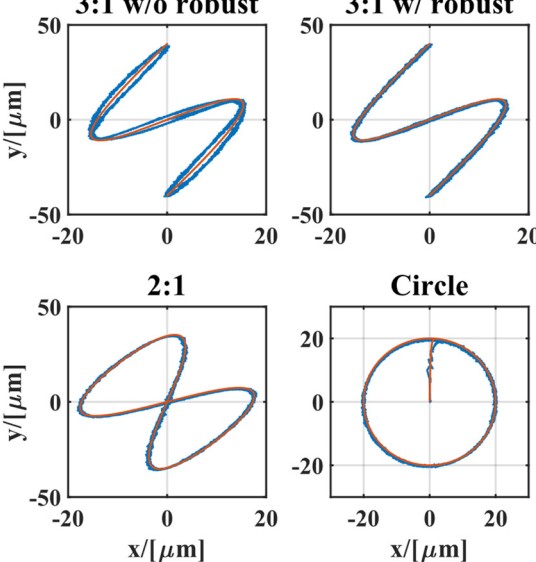

**Figure 11.** Two-dimensional plot of the results from Figures 9 and 10, showing the actual trajectory of the closed-loop CM.

In general, the results shown in Figures 9 and 10 indicate that despite the errors in the physical parameters, i.e., the vibration at the very beginning of the error signals, the controller achieves boundedness and guaranteed transient performance for error signals. As a comparison, as shown in Figure 9a, the non-robust controller suffers from higher noise levels and a larger overall tracking error amplitude. Further, the maximum linear feedback

gain of the non-robust controller is also lower than the robust one due to uncompensated noises and disturbances, resulting in a slower response to fast variating input signals.

On the other hand, however, although the proposed method outperforms the non-robust one to some degree, the deduction for the robust controller is cumbersome and varies in accordance with different system designs. In addition, the nonlinearity in our system is suppressed by a robust controller with knowledge of only a part of the nonlinear deformation denoted by $\delta u$. The whole modeling and controller design process is performed in the corresponding tangent space, which is a major limitation to final performance. Work on larger deformations and more profound nonlinearities remains to be conducted.

## 5. Conclusions

In this paper, we developed an integrated design and control scheme for a large workspace topology-optimized compliant mechanism. The scheme takes the noisy sensors, system distortions, and nonlinearities of the PEAs into consideration. The system model is based on a simplification of a spring–mass system with fully coupled inputs $x_{in}$. The controller described in this study was designed without deliberately trying to decouple the input–output relationship, which is usually unavailable. The proposed controller employs a backstepping procedure with a robust part and an output feedback part. The theoretical analysis was verified through experimental studies.

Future work will be conducted on controller design using full finite-element models or reduced-order modal analysis to achieve better tracking performance. The error analysis mentioned in Section 3.2.1 is also a potential area of future research. The development of an adaptive controller is also a potential research topic, since in the current study, dynamic response and convergence efficiency is limited due to ignorance of the online parameter variations.

**Author Contributions:** Conceptualization, B.J. and W.Z.; Data curation, Y.H.; Formal analysis, Y.H.; Funding acquisition, B.J. and W.Z.; Investigation, W.Z.; Methodology, Y.H.; Project administration, B.J. and W.Z.; Resources, B.J. and W.Z.; Software, Y.H.; Supervision, B.J. and W.Z.; Validation, Y.H. and W.Z.; Visualization, Y.H.; Writing—original draft, Y.H. and B.J.; Writing—review & editing, Y.H. and W.Z. All authors have read and agreed to the published version of the manuscript.

**Funding:** This work was supported by the National Natural Science Foundation of China (NSFC) (No. 52175439), National Natural Science Foundation of Zhejiang Province (No. LD22E050010), National Key R&D Program of China No. 2021YFB3400300, and the Science Fund for Creative Research Groups of the National Natural Science Foundation of China (No. 51821093).

**Institutional Review Board Statement:** Not applicable.

**Informed Consent Statement:** Not applicable.

**Data Availability Statement:** Not applicable.

**Conflicts of Interest:** The authors declare no conflict of interest. The funders had no role in the design of the study; in the collection, analyses, or interpretation of data; in the writing of the manuscript, or in the decision to publish the results.

## Appendix A. Nonlinear Finite-Element Method Synthesis

The spatial discretization of the design domain, with element alignment, node connectivity, and DOF indexing, is shown in Figure A1a, which is in accordance with [44]. In general, there are two main contributors to the nonlinearity of the system design [40]. The first one is geometric nonlinearity, which appears in the form of higher-order terms $H^T H$ in the Green–Lagrange strain tensor, when we take large deformations in the elements into consideration:

$$E = \frac{1}{2}\left(H + H^T + H^T H\right), \tag{A1}$$

where $H = \nabla \bar{u}$, the displacement gradient when the displacement vector $\bar{u}(X, t)$ is introduced. A detailed iso-parametric mapping of the deformation of a finite element, $\Omega_e$, is

depicted in the lower part of Figure A1a: The unit square reference configuration, $\Omega_\square$, is first transformed into the initial configuration, $\Omega_e$, with the coordinates $X_{el}$, $l = 1, 2, 3, 4$ by the Jacobian $J_e$. The initial configuration, $\Omega_e$, is then transformed into the current configuration, $\overline{\varphi}(\Omega_e)$, with coordinates $x_{el}$, $l = 1, 2, 3, 4$ via a deformation gradient $F_e = H_e + I$, where $I$ is the identity matrix.

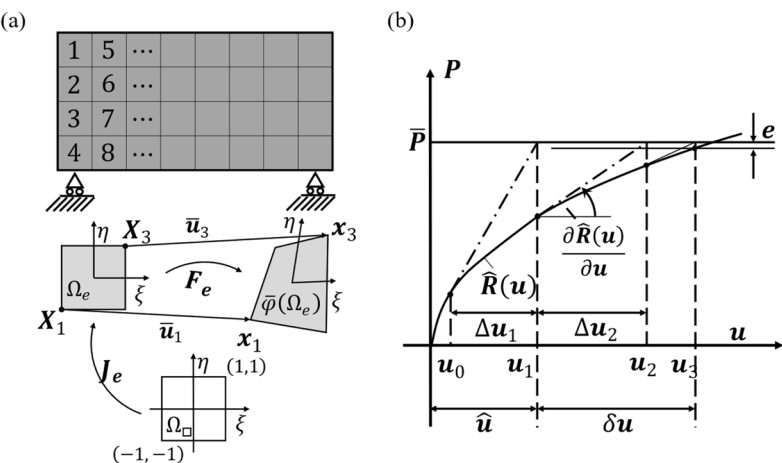

**Figure A1.** (**a**) Schematic illustration of the design domain discretization and elemental deformation; (**b**) Newton–Raphson method for nonlinear finite element analysis.

The second main contributor to the nonlinearity of the system is the material nonlinearity, which occurs when the relation between the stress and strain is not linear. Material nonlinearities are usually neglected in the finite-element analysis of TO. However, in our case, the use of an additive hyper-elastic element will suppress the numerical instability in the low-stiffness region and is thus an effective way to achieve global convergence [41,63]. The basic idea is to add a soft hyper-elastic material with a strain energy function in accordance with the Yeoh model to the low-density elements that are at risk of instability. The additive stiffness energy function of the $e$th element $\Omega_e$ is of the following form:

$$\widetilde{\Psi}_e(I_1) = \left(1 - \rho_e^p\right)\left(c_{1e}(I_1 - 3) + c_{2e}(I_1 - 3)^2\right), \tag{A2}$$

where $I_1 = \text{tr}(\mathfrak{C}_e)$ is the first invariant of the right Cauchy–Green strain tensor $\mathfrak{C}_e$, $\rho_e$ is the elemental density, $p$ is the penalization factor used in the SIMP approach, and $c_{1e} > 0, c_{2e} > 0$ are the material constants of the additive hyper-elastic material for the $e^{\text{th}}$ element. In general, $c_{1e}$ is assumed to be small to sustain a convergent result under little strain, whereas $c_{2e}$ is relatively larger to suppress the instability at larger deformations. These coefficients are updated elementwise in each iteration of the SIMP in accordance with the following [41]:

$$c_{1e} = \frac{\rho_{min}^p E_0}{6}$$
$$c_{2e}^{(k+1)} = \begin{cases} c_{e2}^{(k)}\sqrt{\omega_e^{(k)}}, & \text{if } \eta_e^{(k)} \leq 1 \\ c_{e2}^{(k)}\left(\omega_e^{(k)}\right), & \text{if } \eta_e^{(k)} > 1 \end{cases} \tag{A3}$$
$$\eta_e^{(k)} = \frac{\varepsilon_e^{(k)}}{\varepsilon^\star},$$

where the superscript $(k)$ represents the iteration step of the optimization process, $\varepsilon_e$ is the average von Mises strain of the $e$th element, and $\varepsilon^\star$ is the specified threshold.

The second Piola–Kirchhoff stress $\widetilde{\mathfrak{S}}$ of the additive element is the derivative of the strain energy function with respect to the Gauss–Lagrange strain tensor $\boldsymbol{E}$:

$$\widetilde{\mathfrak{S}}_e = \frac{\partial \widetilde{\Psi}_e}{\partial \boldsymbol{E}} \,\&\, = \frac{2\,\partial \widetilde{\Psi}_e}{\partial \mathfrak{C}} = 2\left(1 - \rho_e^p\right)\left(c_{1e} + 2c_{2e}(I_1 - 3)\right)\frac{\partial I_1}{\partial \mathfrak{C}}. \tag{A4}$$

Notice that the associated nonlinearity of the additive hyper-elastic material only benefits the convergence procedure during the TO iterations and is not included in the actual nonlinear dynamic analysis.

Returning to the spatially discretized design domain with square elements as shown in Figure A1a, the finite-element formulations of the weak form in the initial configuration for the residuum and both boundary loads $\boldsymbol{P}_e^\sigma$ and body force $\boldsymbol{P}_e$ within a single element $\Omega_e$ in the SIMP method are obtained as follows [39]:

$$\begin{aligned}
\boldsymbol{R}_e(\boldsymbol{u}_e, \rho_e, p)| &= \int_{\Omega_e} \boldsymbol{B}(\boldsymbol{u}_e)^T \left(\boldsymbol{S}_e + \widetilde{\mathfrak{S}}_e\right) \det \boldsymbol{J}_e \, \mathrm{d}\square, \\
\boldsymbol{P}_e^\sigma| &= \int_{\Gamma_r} \boldsymbol{N}^T \bar{\boldsymbol{t}} d\Gamma, \\
\boldsymbol{P}_e| &= \int_{\Omega_e} \rho_0 \rho_e \bar{\boldsymbol{b}} \mathrm{d}\Omega,
\end{aligned} \tag{A5}$$

where $\boldsymbol{u}_e : (X, t) \to \mathbb{R}^N$ denotes the displacement vector from the original configuration $\boldsymbol{X}_e$ to the current configuration $\boldsymbol{x}_e$. Further, $\boldsymbol{B}$ denotes the strain–displacement matrix, whereas $\boldsymbol{S}_e = \rho_e \boldsymbol{D} \boldsymbol{E}$ is the Piola–Kirchhoff stress of the original elastic material in the SIMP, and $\boldsymbol{N}$ is the shape function used in the evaluation of the deformations. All variables and matrices are evaluated within element $\Omega_e$ and are related to the initial configuration. The approximated integration is carried out with Gauss integration due to its efficiency.

The assembly operators for the residual forces and mass are the same and denoted by $\cup_e \bullet$, where $\{e\}_{e \in \mathbb{N}^+}$ is the index set for the interior elements. The boundary loads are assembly operators acting on $\Gamma_r \subseteq \partial\Omega$, denoted by $\cup_r \bullet$, where $\{r\}_{r \in \{e\}}$ is the index set for the boundary elements to be considered. The global matrices are expressed as shown below with the assistance of the assembly operators:

$$\begin{aligned}
[\boldsymbol{R}(\boldsymbol{u})]^{2N \times 1} &= \bigcup_{e=1}^{n_e} [\boldsymbol{R}_e(\boldsymbol{u}_e)]^{8 \times 1} \\
[\boldsymbol{P}]^{2N \times 1} &= \bigcup_{e=r}^{n_r} [\boldsymbol{P}_r]^{8 \times 1},
\end{aligned} \tag{A6}$$

where $N$ is the total number of nodes in the discretized design domain. Therefore, $2N$ is the total number of DOF in the two-dimensional setup for all of the nodes, and $n_e$ and $n_r$ are the number of total elements and number of boundary elements, respectively. A detailed assembly algorithm was followed, as presented in previous research [40,64].

The above derivations lead to a compact nonlinear system of ordinary differential equations of the following form:

$$\boldsymbol{R}(\boldsymbol{u}) - \boldsymbol{P} = 0. \tag{A7}$$

We usually denote by $\boldsymbol{K}_T = \frac{\partial \boldsymbol{R}}{\partial \boldsymbol{u}}\Big|_{\boldsymbol{u}} = \cup_{e=1}^{n_e} \boldsymbol{K}_{eT}$ the assembly of the elementary tangent stiffness matrix $\boldsymbol{K}_{eT} = \frac{\partial \boldsymbol{R}_e(\boldsymbol{u}_e)}{\partial \boldsymbol{u}_e}\Big|_{\boldsymbol{u}_e}$ at the state $\boldsymbol{u}(\boldsymbol{u}_e)$, as shown in Figure A1b. The approximated solution $\boldsymbol{u}$ of the nonlinear dynamic system in Equation (A7) is obtained via a modified Newton–Raphson method.

## Appendix B. Identifying the Hysteresis Model of the PEAs

Various previous studies have reported on the identification procedure for the PEA coefficients [2,48,56]. In this study, we used recursive least-squares estimation to estimate the model parameters. We rewrote Equation (18) in the form of a state-space representation. Suppose that $\hat{x}_{nom}$ represents the predicted output vector of the model. By selecting the parameter vector $\boldsymbol{\theta} = \begin{bmatrix} \alpha & B_{bw} & \Gamma_{bw} & \delta_{bw} \end{bmatrix}^T$, where $\alpha = d_p - A_{bw}$ is a combined

intermediate parameter, and the state variables $\boldsymbol{\varphi} = \begin{bmatrix} \dot{U} & |\dot{U}|h & \dot{U}|h| & -U\text{sgn}(\dot{U}) \end{bmatrix}^T$, a time derivative of the first formula in Equation (18) yields the following:

$$\hat{\dot{x}}_{nom}(t, \boldsymbol{\theta}) = \boldsymbol{\varphi}(t)^T \boldsymbol{\theta}. \tag{A8}$$

A covariance matrix for the recursive least square estimation process can be given as follows [27,64]:

$$\boldsymbol{Cov}(t) = \left[ \sum_{i=1}^{t} \boldsymbol{\varphi}(i) \boldsymbol{W}(t, i) \boldsymbol{\varphi}(i)^T \right]^{-1} \in \mathbb{R}^{p \times p}. \tag{A9}$$

where $\boldsymbol{W}(t, i)$ represents the weighting matrices, in the form of:

$$\boldsymbol{W}(t, n) = \prod_{m=n+1}^{t} \lambda_f(m) \boldsymbol{W}_0. \tag{A10}$$

The $\lambda_f(k)$ in the above expression is the forgetting factor, applied to improve the real-time performance against system disturbances, and $\boldsymbol{W}_0 \in \mathbb{R}^{m \times m}$ is an arbitrary constant weighting matrix. We obtain the following recursive formula for least-squares estimation based on the parameter estimate $\hat{\boldsymbol{\theta}}(t-1)$ and covariance matrix $\boldsymbol{Cov}(t-1)$ obtained in the previous step:

$$
\begin{aligned}
\boldsymbol{\Gamma}(t) &= \tfrac{1}{\lambda_f(t)} \boldsymbol{Cov}(t-1) \\
\boldsymbol{Cov}(t) &= \boldsymbol{\Gamma}(t) \{ I_p - \boldsymbol{\varphi}(t) [\boldsymbol{W}_0^{-1} + \boldsymbol{\varphi}(t) \boldsymbol{\Gamma}(t) \boldsymbol{\varphi}(t)]^{-1} \boldsymbol{\varphi}(t)^T \boldsymbol{\Gamma}(t) \} \\
\epsilon^0(t) &:= \hat{\dot{x}}_{nom}(t) - \boldsymbol{\varphi}(t)^T \hat{\boldsymbol{\theta}}(t-1) \\
\hat{\boldsymbol{\theta}}(t) &= \hat{\boldsymbol{\theta}}(t-1) + \boldsymbol{Cov}(t) \boldsymbol{\varphi}(t) W_0 \epsilon^0(t).
\end{aligned} \tag{A11}
$$

The intermediate adaptation rate matrix $\boldsymbol{\Gamma}(t)$ is introduced for notational simplicity and efficiency of computation. $\epsilon^0$ is normally called the *a priori* prediction error. The $d_p$ in the combined intermediate variable $\alpha$ is estimated by a simultaneous process with a structure identical to the one given in Equation (A11), where $\varphi(t) = U(t)$ and $h(t)$ is updated via a forward Euler algorithm, in accordance with the first formula in Equation (18). Choose another set of parameter vector $\boldsymbol{\varphi}_{nom} = \begin{bmatrix} d_p & -1 \end{bmatrix}^T$ and state variable vector $\boldsymbol{\theta}_{nom} = \begin{bmatrix} U & h \end{bmatrix}$, the parallel RLSE process can be formulated as follows:

$$\hat{x}_{nom}(t, \boldsymbol{\theta}) = \boldsymbol{\varphi}_{nom}(t)^T \boldsymbol{\theta}_{nom}. \tag{A12}$$

Thus, by fixing the constant $-1$ in the $\boldsymbol{\varphi}_{nom}$, the value of $d_p$ can be estimated via an integrated process based on $U$ and $h$. Meanwhile, the value of the parameter $A_{bw} = d_p - \alpha$ can also be calculated at every time step of the identification procedure. Details of the algorithm are given in Algorithm A1, and the estimation results are given in Figure A2 and Table A1.

---

**Algorithm A1 Parallel recursive least square algorithm for online parameter estimation.**
**Parallel RLSE Algorithm**

---

**Initialization:**
**Make an initial guess of the coefficients, compute corresponding initial values of θ and φ**
**Main loop:**
**While PEA is working do**
**Do process (A11) for regressor Equation (A8)**
**Update state variable h**
**Do process (A11) for regressor Equation (A12)**
**Update $d_p$ and $A_{bw}$**
**End while**

---

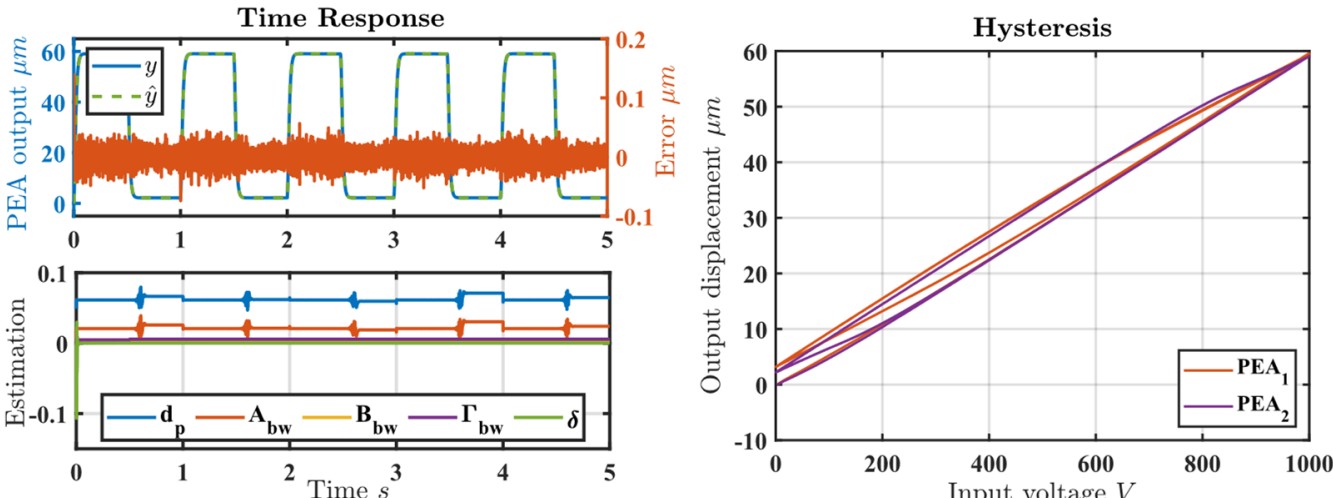

**Figure A2.** Parameter identification process for one of the PEAs and the corresponding hysteresis curve for both.

**Table A1.** Estimation results of characteristic coefficients of the PEAs.

| Symbol | Initial Guess | Estimates | |
| --- | --- | --- | --- |
| | | PEA1 | PEA2 |
| $d_p$ | 0 | $\approx 6.2 \times 10^{-2}$ | $\approx 6.5 \times 10^{-2}$ |
| $A_{bw}$ | 0 | $\approx 1.2 \times 10^{-2}$ | $\approx 2.5 \times 10^{-2}$ |
| $B_{bw}$ | 0 | $1.9 \times 10^{-3}$ | $1.8 \times 10^{-3}$ |
| $\Gamma_{bw}$ | 0 | $1.6 \times 10^{-3}$ | $5.3 \times 10^{-3}$ |
| $\delta$ | 0 | $5.015 \times 10^{-4}$ | $1.973 \times 10^{-4}$ |

**Appendix C. The State-Space Representation of the System Dynamics**

Following the steps given in Section 3.2.1, the elementary form of the total system dynamics can be given as follows:

$$
\begin{aligned}
\dot{z}_1 &= z_5 \\
\dot{z}_2 &= z_6 \\
\dot{z}_3 &= z_7 \\
\dot{z}_4 &= z_8 \\
\dot{z}_5 &= \sum_{i=1}^{8} \beta_{5i} z_i \\
\dot{z}_6 &= \sum_{i=1}^{8} \beta_{6i} z_i \\
\dot{z}_7 &= d_{11} u_1 + d_{12} u_2 + \sum_{i=1}^{8} \beta_{7i} z_i \\
\dot{z}_8 &= d_{21} u_2 + d_{22} u_2 + \sum_{i=1}^{8} \beta_{8i} z_i
\end{aligned}
\tag{A13}
$$

This linear differential equation system can be modified into the state-space form as Equation (36), whose coefficient matrices $B = [\beta_{ij}]^{8\times8}$ and $D = [d_{ij}]^{2\times2}$ are detailed below:

$$
\begin{bmatrix} \beta_{51} & \beta_{52} \\ \beta_{61} & \beta_{62} \end{bmatrix} = \begin{bmatrix} -\frac{2K_{qo1}}{M} & 0 \\ 0 & -\frac{K_{qo2}}{M} \end{bmatrix},\;
\begin{bmatrix} \beta_{53} & \beta_{54} \\ \beta_{63} & \beta_{64} \end{bmatrix} = \begin{bmatrix} \frac{K_{qo1}}{M} & \frac{K_{qo1}}{M} \\ \frac{\alpha K_{qo2}}{M} & -\frac{\alpha K_{qo2}}{M} \end{bmatrix},
$$

$$
\begin{bmatrix} \beta_{55} & \beta_{56} \\ \beta_{65} & \beta_{66} \end{bmatrix} = \begin{bmatrix} -\frac{2C_{qo1}}{M} & 0 \\ 0 & -\frac{C_{qo2}}{M} \end{bmatrix},\;
\begin{bmatrix} \beta_{57} & \beta_{58} \\ \beta_{67} & \beta_{68} \end{bmatrix} = \begin{bmatrix} \frac{C_{qo1}}{M} & \frac{C_{qo1}}{M} \\ \frac{\alpha C_{qo2}}{M} & -\frac{\alpha C_{qo2}}{M} \end{bmatrix},
$$

$$
\begin{bmatrix} \beta_{71} & \beta_{72} \\ \beta_{81} & \beta_{82} \end{bmatrix} = \begin{bmatrix} \frac{K_{qo1}}{M_{in1}} & 0 \\ \frac{K_{qo1}}{M_{in2}} & 0 \end{bmatrix},
$$

$$
\begin{bmatrix} \beta_{73} & \beta_{74} \\ \beta_{83} & \beta_{84} \end{bmatrix} = \begin{bmatrix} -\frac{K_{in1}+K_{pea1}+K_{qo1}}{M_{in1}} & 0 \\ 0 & -\frac{K_{in2}+K_{pea2}+K_{qo1}}{M_{in2}} \end{bmatrix},
$$

$$
\begin{bmatrix} \beta_{75} & \beta_{76} \\ \beta_{85} & \beta_{86} \end{bmatrix} = \begin{bmatrix} \frac{C_{qo1}}{M_{in1}} & 0 \\ \frac{C_{qo1}}{M_{in2}} & 0 \end{bmatrix},\;
\begin{bmatrix} \beta_{77} & \beta_{78} \\ \beta_{87} & \beta_{88} \end{bmatrix} = \begin{bmatrix} -\frac{C_{in1}+C_{qo1}}{M_{in1}} & 0 \\ 0 & -\frac{C_{in2}+C_{qo1}}{M_{in2}} \end{bmatrix},
$$

$$
\begin{bmatrix} d_{11} & d_{12} \\ d_{21} & d_{22} \end{bmatrix} = \begin{bmatrix} \frac{K_{pea1}}{M_{in1}} & 0 \\ 0 & \frac{K_{pea2}}{M_{in2}} \end{bmatrix}.
\tag{A14}
$$

## Appendix D. Deduction of the Backstepping Controller

Recall that $a(\eta) = [a_1, a_2]^T$ represents the ideal control law for the inputs $[x_{in1}, x_{in2}]^T$ according to Equation (A13), the notation that $z = \begin{bmatrix} q_o & \dot{q}_o & x_{in} & \dot{x}_{in} \end{bmatrix}^T$, and the subsystem Equation (43a) in terms of $a$ is given as follows:

$$
\begin{aligned}
\dot{\eta}_1 &= \eta_5 := a_1 \\
\dot{\eta}_2 &= \eta_6 := a_2 \\
\dot{\eta}_3 &= \dot{\xi}_1 \\
\dot{\eta}_4 &= \dot{\xi}_2 \\
\dot{\eta}_5 &= \sum_i^6 \beta_{5i}\eta_i + \beta_{57}\dot{\xi}_1 + \beta_{58}\dot{\xi}_2 + \Delta_{\eta1} \\
\dot{\eta}_6 &= \sum_i^6 \beta_{6i}\eta_i + \beta_{67}\dot{\xi}_1 + \beta_{68}\dot{\xi}_2 + \Delta_{\eta2}
\end{aligned}
\tag{A15}
$$

When denoting $\epsilon_i = \eta_i - \eta_{di} = z_i - z_{di}$, $i = 1,2$, the tracking error between the output node $\eta_i = z_i$, and the desired trajectory $\eta_{di} = z_{di}$, a selection of the feedback error $s = \dot{\epsilon} + k_\epsilon \epsilon$ yields the total error dynamic:

$$
\dot{s}_1 = -\frac{2K_{qo1}}{M}\eta_1 + \frac{K_{qo1}}{M}a_1 + \frac{K_{qo1}}{M}a_2 + \beta_{57}\dot{\xi}_1 + \beta_{58}\dot{\xi}_2 - \ddot{\eta}_{d1} + k_{\epsilon1}(\eta_5 - \dot{\eta}_{d1}) + \Delta_{\eta1}, \tag{A16}
$$

$$
\dot{s}_2 = -\frac{K_{qo2}}{M}\eta_2 + \alpha\frac{K_{qo2}}{M}a_1 - \alpha\frac{K_{qo2}}{M}a_2 + \beta_{67}\dot{\xi}_1 + \beta_{68}\dot{\xi}_2 - \ddot{\eta}_{d2} + k_{\epsilon2}(\eta_6 - \dot{\eta}_{d2}) + \Delta_{\eta2}. \tag{A17}
$$

Consider the control Lyapunov function (CLF) Equations (47) and (48):

$$
V(\eta) = \frac{1}{2}\begin{bmatrix} s_1^2 \\ s_2^2 \end{bmatrix}.
\tag{A18}
$$

The path derivative of the CLF with respect to the solution $\eta(t)$ is as follows:

$$\dot{V}(\eta) = \begin{bmatrix} s_1 \dot{s}_1 \\ s_2 \dot{s}_2 \end{bmatrix}. \tag{A19}$$

Next, we choose a specific $a$ such that $\dot{V}(\eta) < 0$, satisfying the Lyapunov stability criterion. Substituting Equations (A16) and (A17) into Equation (A19) and considering the inequalities, we have

$$\begin{cases} s_1\left(\frac{K_{qo1}}{M}(a_1 + a_2 - 2\eta_1) - \ddot{\eta}_{d1} + k_{\epsilon 1}(\eta_5 - \dot{\eta}_{d1}) + (\beta_{57}\hat{\dot{\xi}}_1 + \beta_{58}\hat{\dot{\xi}}_2) + \Delta_{\eta 1}\right) < 0 \\ s_2\left(\frac{\alpha K_{qo2}}{M}(a_1 - a_2 - \frac{\eta_2}{\alpha}) - \ddot{\eta}_{d2} + k_{\epsilon 2}(\eta_6 - \dot{\eta}_{d2}) + (\beta_{67}\hat{\dot{\xi}}_1 + \beta_{68}\hat{\dot{\xi}}_2) + \Delta_{\eta 2}\right) < 0 \end{cases}. \tag{A20}$$

The desired controller output $a$ as a combination of $a_m$ and $a_s$ from Equation (49) satisfies the following:

$$\begin{cases} \frac{K_{qo1}}{M}(a_{m1} + a_{m2}) - Z_5\left(\eta_1, \dot{\eta}_1, \hat{\dot{\xi}}_1\right) = -(k_{\eta 1} - 1)s_1 \\ \frac{\alpha K_{qo2}}{M}(a_{m1} - a_{m2}) - Z_6\left(\eta_1, \dot{\eta}_1, \hat{\dot{\xi}}_1\right) = -(k_{\eta 2} - 1)s_2 \end{cases}, \tag{A21}$$

$$\begin{cases} \frac{K_{qo1}}{M}(a_{m1} + a_{m2}) + \Delta_{\eta 1} < \frac{K_{qo1}}{M}(a_{m1} + a_{m2}) + \max|\Delta_{\eta 1}| = -s_1 \\ \frac{\alpha K_{qo2}}{M}(a_{m1} - a_{m2}) + \Delta_{\eta 2} < \frac{\alpha K_{qo2}}{M}(a_{m1} - a_{m2}) + \max|\Delta_{\eta 2}| = -s_2 \end{cases}, \tag{A22}$$

where $k_{\eta i} > 1, i = 1, 2$ are the linear feedback gains and can be chosen freely, and

$$\begin{aligned} Z_5\left(\eta_1, \dot{\eta}_1, \dot{\xi}_1\right) &= \frac{2K_{qo1}}{M}\eta_1 + \left(\ddot{\eta}_{d1} - k_{\epsilon 1}(\eta_5 - \dot{\eta}_{d1})\right) - \left(\beta_{57}\hat{\dot{\xi}}_1 + \beta_{58}\hat{\dot{\xi}}_2\right), \\ Z_6\left(\eta_2, \dot{\eta}_2, \dot{\xi}_2\right) \& &= \frac{K_{qo2}}{M}\eta_2 + \left(\ddot{\eta}_{d2} - k_{\epsilon 2}(\eta_6 - \dot{\eta}_{d2})\right) - \left(\beta_{67}\hat{\dot{\xi}}_1 + \beta_{68}\hat{\dot{\xi}}_2\right). \end{aligned} \tag{A23}$$

By solving Equations (A21) and (A22), we get the following expressions for the ideal control law $a$:

$$a_m = \begin{bmatrix} K_{qo1}/M & K_{qo1}/M \\ \alpha K_{qo2}/M & -\alpha K_{qo2}/M \end{bmatrix}^{-1} \left( \begin{bmatrix} -(k_{\eta 1} - 1)s_1 + Z_5 \\ -(k_{\eta 2} - 1)s_2 + Z_6 \end{bmatrix} \right), \tag{A24}$$

$$a_s = \begin{bmatrix} K_{qo1}/M & K_{qo1}/M \\ \alpha K_{qo2}/M & -\alpha K_{qo2}/M \end{bmatrix}^{-1} \begin{bmatrix} -\max|\Delta_{\eta 1}| - s_1 \\ -\max|\Delta_{\eta 2}| - s_2 \end{bmatrix}. \tag{A25}$$

Let $\epsilon_a$ be the deviation from Equation (50):

$$\epsilon_a = \xi - a. \tag{A26}$$

The total system in the new error coordinates $\begin{bmatrix} s & \epsilon_a \end{bmatrix}^T$ is then given by:

$$\begin{aligned} \dot{s} &= -\begin{bmatrix} k_{\eta 1}s_1 \\ k_{\eta 2}s_2 \end{bmatrix} \\ \dot{\epsilon}_a &= v - \begin{bmatrix} K_{qo1}/M & K_{qo1}/M \\ \alpha K_{qo2}/M & -\alpha K_{qo2}/M \end{bmatrix}^{-1} \left( \begin{bmatrix} k_{\eta 1}(k_{\eta 1} - 1)s_1 + \frac{\partial Z_5(\eta)}{\partial \eta} \\ k_{\eta 2}(k_{\eta 2} - 1)s_2 + \frac{\partial Z_6(\eta)}{\partial \eta} \end{bmatrix} \right) + \Delta_\epsilon, \end{aligned} \tag{A27}$$

where the error term $\Delta_\epsilon$ is derived from Equation (43b), such that $\Delta_\epsilon = \Delta_\xi$.

The CLF for the whole system in accordance with Equation (51) can be given as follows:

$$V_\alpha(\boldsymbol{\eta}, \boldsymbol{\xi}) = V(\boldsymbol{\eta}) + \frac{1}{2}\begin{bmatrix} \epsilon_{\alpha 1}^2 \\ \epsilon_{\alpha 2}^2 \end{bmatrix}. \tag{A28}$$

The path derivative of the above Equation (A28) along the solutions of Equation (43a) can be calculated by substituting the expression of $\dot{s}$ in Equation (A27) into Equation (52), in the following form:

$$\dot{V}_\alpha = \begin{bmatrix} -k_{\eta 1}s_1^2 \\ -k_{\eta 2}s_2^2 \end{bmatrix} + \begin{bmatrix} \epsilon_{\alpha 1}\dot{\epsilon}_{\alpha 1} \\ \epsilon_{\alpha 2}\dot{\epsilon}_{\alpha 2} \end{bmatrix}. \tag{A29}$$

The term $\epsilon_\alpha \dot{\epsilon}_\alpha$ can further be written as a linear combination of input $v$. Denoting $\begin{bmatrix} \theta_1 & \theta_2 \end{bmatrix} = \begin{bmatrix} \frac{K_{qo1}}{M} & \frac{K_{qo2}}{M} \end{bmatrix}$, we have the following total system formulation for Equation (A29):

$$\begin{aligned}
\dot{V}_{\alpha 1} &= -k_{\eta 1}s_1^2 + \left( \hat{\dot{\xi}}_1 + \Pi_{11} \right)(v_1 - \Pi_{12}) + \epsilon_{\alpha 1}\Delta_{\epsilon 1} \\
\dot{V}_{\alpha 2} &= -k_{\eta 2}s_2^2 + \left( \hat{\dot{\xi}}_2 + \Pi_{21} \right)(v_2 - \Pi_{22}) + \epsilon_{\alpha 1}\Delta_{\epsilon 2}
\end{aligned}, \tag{A30}$$

where

$$\begin{aligned}
\Pi_{11} &= \frac{\left(\max|\Delta_{\eta 2}| - Z_6 + k_{\eta 2}s_2\right)\theta_1 + \alpha\left(\max|\Delta_{\eta 1}| - Z_5 + k_{\eta 1}s_1\right)\theta_2}{2\alpha\theta_1\theta_2} \\
\Pi_{12} &= \frac{\left(k_{\eta 2}\left(k_{\eta 2}-1\right)s_2 + \frac{\partial Z_6(\eta)}{\partial \eta}\right)\theta_1 + \alpha\left(s_1 k_{\eta 1}\left(k_{\eta 1}-1\right) + \frac{\partial Z_5(\eta)}{\partial \eta}\right)\theta_2}{2\alpha\theta_1\theta_2}
\end{aligned}, \tag{A31}$$

$$\begin{aligned}
\Pi_{21} &= -\frac{\left(\max|\Delta_{\eta 2}| - Z_6 + k_{\eta 2}s_2\right)\theta_1 - \alpha\left(\max|\Delta_{\eta 1}| - Z_5\theta_2 + k_{\eta 1}s_1\right)\theta_2}{2\alpha\theta_1\theta_2} \\
\Pi_{22} &= \frac{\left(k_{\eta 2}\left(k_{\eta 2}-1\right)s_2 + \frac{\partial Z_6(\eta)}{\partial \eta}\right)\theta_1 - \alpha\left(k_{\eta 1}\left(k_{\eta 1}-1\right)s_1 + \frac{\partial Z_5(\eta)}{\partial \eta}\right)\theta_2}{2\alpha\theta_1\theta_2}
\end{aligned}. \tag{A32}$$

The task is to choose inputs $v_i$s such that $\dot{V}_{\alpha i}$, $i = 1, 2$ are negative definite, in which case the whole system is Lyapunov stable. Suppose that our controller input is of the following form:

$$v_m + v_s = \begin{bmatrix} v_{m1} + v_{s1} \\ v_{m2} + v_{s2} \end{bmatrix}, \tag{A33}$$

and notice that $-k_{\eta i}s_i^2 \leq 0$ always holds. The values of $v_{mi}$s are then chosen such that:

$$\begin{cases} \left( \hat{\dot{\xi}}_1 + \Pi_{11} \right)(v_{m1} - \Pi_{12}) = -k_1\epsilon_{a1} \\ \left( \hat{\dot{\xi}}_2 + \Pi_{21} \right)(v_{m2} - \Pi_{22}) = -k_2\epsilon_{a2} \end{cases}, \tag{A34}$$

where $\boldsymbol{k} = \begin{bmatrix} k_1 & k_2 \end{bmatrix} > 0$ is the feedback gain, thus yielding:

$$\begin{cases} v_{m1} = -\frac{k_1\epsilon_{a1}}{\hat{\dot{\xi}}_1 + \Pi_{11}} + \Pi_{12} \\ v_{m2} = -\frac{k_2\epsilon_{a2}}{\hat{\dot{\xi}}_2 + \Pi_{21}} + \Pi_{22} \end{cases}. \tag{A35}$$

The system nonlinearity is overcome by the robust controller $v_{si}$s, satisfying the following relationships:

$$\begin{cases} \left( \hat{\dot{\xi}}_1 + \Pi_{11} \right)v_{s1} + \epsilon_{\alpha 1}\Delta_{\epsilon 1} < (\hat{\dot{\xi}}_1 + \Pi_{11})v_{s1} + \max|\epsilon_{\alpha 1}\Delta_{\epsilon 1}| = -\epsilon_{\alpha 1} \\ \left( \hat{\dot{\xi}}_2 + \Pi_{21} \right)v_{s2} + \epsilon_{\alpha 2}\Delta_{\epsilon 2} < (\hat{\dot{\xi}}_2 + \Pi_{21})v_{s2} + \max|\epsilon_{\alpha 2}\Delta_{\epsilon 2}| = -\epsilon_{\alpha 2} \end{cases}, \tag{A36}$$

and the expression of the $v_{si}$s can be solved as follows:

$$v_{s1} = -\frac{\max|\epsilon_{\alpha 1}\Delta_{\epsilon 1}| + \epsilon_{\alpha 1}}{\hat{\dot{\zeta}}_1 + \Pi_{11}}$$
$$v_{s2} = -\frac{\max|\epsilon_{\alpha 2}\Delta_{\epsilon 2}| + \epsilon_{\alpha 2}}{\hat{\dot{\zeta}}_2 + \Pi_{21}}$$
$$\text{(A37)}$$

We can also choose identical feedback gains such that $k_i = k_{mi} = k$. The final design of the controller input, Equation (A33), as a combination of Equations (A35) and (A37), is simplified to:

$$v_1 = -\frac{k_1\epsilon_{a1}}{\hat{\dot{\zeta}}_1 + \Pi_{11}} - \frac{\max|\epsilon_{\alpha 1}\Delta_{\epsilon 1}| + \epsilon_{\alpha 1}}{\hat{\dot{\zeta}}_1 + \Pi_{11}} + \Pi_{12}$$
$$v_2 = -\frac{k_2\epsilon_{a2}}{\hat{\dot{\zeta}}_2 + \Pi_{21}} - \frac{\max|\epsilon_{\alpha 2}\Delta_{\epsilon 2}| + \epsilon_{\alpha 2}}{\hat{\dot{\zeta}}_2 + \Pi_{21}} + \Pi_{22}$$
$$\text{(A38)}$$

Also, all the coefficients are calculated with respect to estimation of the state varia.

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
