# Peer review of "Integrated Development of a Topology-Optimized Compliant Mechanism for Precise Positioning"

_actuators, doi:10.3390/act11070179_

Round 1

Reviewer 1 Report

This is a well presented work which covers well the methodology used to design and implement a controller for a 2D compliant mechanism. The assumptions and optimization methods are explained sufficiently for implementation by others along with the references which provide the foundations. The experimentation and measurements are sufficient to verify the improved operation using the demonstrated techniques.

My question is the assembly limitations you mentioned. How would including the assembly joints affect the model with increased compliance and damping (transitioning Non-newtonian materials) in the total mechanism?  

Perhaps it would also be enlightening to see what the response of the non-linear controller does when the controller is completely linear and when the non-linear parameters are substantially improper. (One would expect an increase in errors over the linear non-robust case).

Author Response

Thank you for the comment. Including the assembly joints will increase the system damping and introduce more nonlinearities because of impact and energy loss at the joint interface from an intuitive point of view. More detailed inspection of such mechanisms still need further research efforts, either in developing more robust controller or in unveiling more details in the system dynamics to persue better system performances.

Reviewer 2 Report

uthors propose an integrated design and control scheme for a large workspace topology-optimized compliant mechanism driven by piezoelectic actuators.
The first two paragraphs (Introduction and Topology Optimization) presents sufficient information regarding the topic addressed in the paper and also about how the authors structured their research. The equations and mathematical models presented in the paper seem to be correct. Overall the work is well structured and presented but there are some aspects that need to be mentioned:

- Reference to equations are missing at lines 363, 630, 631
- The authors use two capacitive distance sensors to measure the movement at the end of the compliant mechanism in accordance with the required trajectory. But one of the sensors is aimed at an element that has a compliant joint at one end. The accuracy of the measurement is not influenced by the deformation at the level of the compliant joint?
- To my knowledge, the Bouc Wen model for hysteresis compensation is mainly used in open loop control systems. In the sense that there is no information regarding the movement of the actuator. The authors of the paper mention that the movement at the end of the compliant mechanism is performed with 2 capacitive sensors. Also from here the speed is estimated. Taking into account the fact that the measurements are made after the actuator and the compliant mechanism, what is the purpose of implementing a Bouc-Wen algorithm? Because all the nonlinearities are before the measurement system. This should probably be specified somewhere in the paper.
- Also, the whole control algorithm is quite complex and requires quite a lot of calculations. Is this really necessary? If so, what are the reasons that justify the implementation of such an algorithm in a practical system

In my opinion the paper presented is valuable but needs some improvements before it can be published.

Author Response

Thank you for the comment. Please see the attachment for point to point response.

Reviewer 3 Report

Please follow the attached remarks. Particularly important is remark No. 8. 

Additionally, please consider to  rewrite Conclusions section. In the current form it is rather Summary.

Author Response

Thank you for the comment. Most of the problems listed in the notes are addressed in the revised version of the manuscript. The abstract is modified to more precisely cover the points of system and method considered in the article. We apologize that due to time and budget limitations we are not able to carry out more experiments, i.e., the proposed frequency response experiment, and that although, in general, frequency tests are not vital for predictive controllers since their response are "model-based", a frequency response experiment would definitely carry more information about the performance of the system. We have modified the manuscript to some degree and some of the responses to similiar questions can also be found in replies to the 2nd reviewer.
